# Meta Internal Learning

**Raphael Bensadoun**[*]
The School of Computer Science
Tel Aviv University

**Shir Gur**
The School of Computer Science
Tel Aviv University

**Tomer Galanti**
The School of Computer Science
Tel Aviv University

**Lior Wolf**
The School of Computer Science
Tel Aviv University

## Abstract

Internal learning for single-image generation is a framework where a generator is trained to produce novel images based on a single image. Since these models are trained on a single image, they are limited in their scale and application. To overcome these issues, we propose a meta-learning approach that enables training over a collection of images, in order to model the internal statistics of the sample image more effectively. In the presented meta-learning approach, a single-image GAN model is generated given an input image, via a convolutional feedforward hypernetwork $f$. This network is trained over a dataset of images, allowing for feature sharing among different models and for interpolation in the space of generative models. The generated single-image model contains a hierarchy of multiple generators and discriminators. Therefore, the meta-learner needs to be trained in an adversarial manner, which requires careful design choices that we justify by a theoretical analysis. Our results show that the models obtained are as suitable as single-image GANs for many common image applications, and significantly reduce training time per image, without loss in performance, and introduce novel capabilities, such as interpolation and feedforward modeling of novel images. Our code is available at: https://github.com/RaphaelBensTAU/MetaInternalLearning.

## 1 Introduction

In the field of internal learning, one wishes to learn the internal statistics of a signal in order to perform various downstream tasks. In this work, we focus on Single image GANs [31, 32, 13, 8], which present extremely impressive results in modeling the distribution of images that are similar to the input image, and in applying this distribution to a variety of applications. However, given that there is no shortage of unlabeled images, one may ask whether a better approach would be to model multiple images, and only then condition the model on a single input image. Doing so, one could (i) benefit from knowledge and feature sharing between the different images, (ii) better define the boundaries between the distribution obtained from the input image and those of other images, (iii) possibly avoid the costly training phase for a novel image, and instead employ feedforward inference, and (iv) mix different single-image models to create novel types of images.

---

[*]Corresponding author - bensadoun@mail.tau.ac.il

35th Conference on Neural Information Processing Systems (NeurIPS 2021).

From the algorithmic standpoint, this multi-image capability can be attempted using various forms of conditioning. For example, one can add a one-hot vector as an input, or, more generally, a vector signature, and train multiple images using the same single image method. One can also add a complete layer of a conditioning signal to the RGB input. Alternatively, one can employ StyleGAN-like conditioning and modify the normalization of the layers [16]. More generally, observing that this scenario is a meta-learning problem, one can employ methods such as MAML [5] for learning a central network and its variants per image. After performing many such attempts over a long period of time, we were unable to achieve a desirable level of performance with any of these methods.

Instead, we advocate for a meta-learning solution that is based on hypernetworks [9]. Hypernetworks consist of two main components: a primary network $g$ that performs the actual computation, and the hypernetwork $f$ that is used for conditioning. The parameters (weights) of $g$ are not learned conventionally. Instead, they are given as the output of $f$ for the conditioned input signal. Following a single-image GAN setting with a hierarchical structure, we have two hypernetworks $f_g$ and $f_d$, which produce the weights of the multiple generators and discriminators for input image $I$ dynamically.

Our method allows for training on multiple images at once, obtaining similar results for various applications to those previously demonstrated for single-image training. It also allows us to interpolate between single-image GANs derived from pairs (or more) of images. Finally, we are able to fit a new unseen image in a fraction of the time required for training a new single-image GAN, *i.e.* our method enables inference generation for a novel image.

As far as we can ascertain, ours is the first method to perform adversarial training using hypernetworks. We provide a theoretical analysis of the proper way to perform this. This analysis demonstrates both the sufficiency of our algorithm for minimizing the objective function and the necessity of various components in our method.

## 2  Background

In this paper we consider the meta-learning problem of learning how to generate a variety of samples from a single image, where each individual learning problem is defined by this single input image. For this purpose, we first recall the setting of single-image generation as in [31, 13, 8].

### 2.1  Single-Image Generation

SinGAN [31] is composed of a multi-scale residual generator $G = \{g_1, \ldots, g_k\}$ and a patch-discriminator $D = \{d_1, \ldots, d_k\}$, where $g_i$ and $d_i$ are fully-convolutional networks, consisting of five layers and used for training at scale $i$. Given an image $I$, we pre-compute $k$ scales of the image, from coarsest to finest, denoted by $I_i$, with height and width $h_i$ and $w_i$, and use each $I_i$ for training the $i$'th generator $g_i$.

The first generator $g_1$ takes as input fixed random noise $z_1 \in \mathbb{R}^{3 \times h_1 \times w_1}$ whose coordinates are i.i.d. normally distributed, and outputs an image $\hat{I}_1 \in \mathbb{R}^{3 \times h_1 \times w_1}$. Every other generator $g_i$ takes as input an upsampled version $\hat{I}_{i-1}^{\uparrow}$ of the previous output $\hat{I}_{i-1}$, plus noise $z_i \in \mathbb{R}^{3 \times h_i \times w_i}$ (whose coordinates are i.i.d. normally distributed), and recursively generates a sample at scale $i$ as follows:

$$\hat{I}_1 := g_1(z_1) := \hat{g}_1(z_1), \quad \hat{I}_i := g_i(\hat{I}_{i-1}, z_i) := \hat{I}_{i-1}^{\uparrow} + \hat{g}_i(\hat{I}_{i-1}^{\uparrow} + z_i), \quad i > 1 \qquad (1)$$

For each scale $i \in [k]$, we denote by $\mathcal{D}_{I,i}$ the distribution of patches $u_{I,i}$ within $I_i$ and by $\mathcal{D}_{\hat{I},i}$ the distribution of patches $u_{\hat{I},i}$ within $\hat{I}_i$ (for $z_1, \ldots, z_i \sim \mathcal{N}(0, \mathbb{I})$). The goal of this method is to train each generator $g_i$ to generate samples $\hat{I}_i$, such that, $\mathcal{D}_{\hat{I},i}$ and $\mathcal{D}_{I,i}$ would closely match.

For this task, the generators $g_i$ are progressively optimized to minimize the 1-Wasserstein distance $W(\mathcal{D}_{\hat{I},i}, \mathcal{D}_{I,i})$ between the distributions $\mathcal{D}_{\hat{I},i}$ and $\mathcal{D}_{I,i}$. The 1-Wasserstein distance between two distributions $\mathcal{D}_1$ and $\mathcal{D}_2$ is defined as follows:

$$W(\mathcal{D}_1, \mathcal{D}_2) := \max_{d: \|d\|_L \leq 1} \left\{ \mathop{\mathbb{E}}_{u \sim \mathcal{D}_1} d(u) - \mathop{\mathbb{E}}_{u \sim \mathcal{D}_2} d(u) \right\}, \qquad (2)$$

where $\|d\|_L$ is the Lipschitz norm of the discriminator $d$.

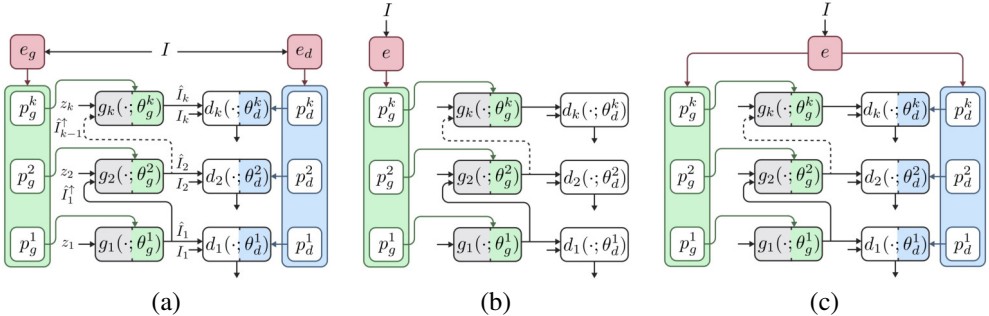

Figure 1: **Alternative architectures for hypernetwork single image generators. (a)** Our model architecture consists of two embedding networks - $e_g$ for the *hyper-generator* and $e_d$ for the *hyper-discriminator*. The primary networks $g_i$ and $d_i$ follow Sec. 2. **(b)** Hyper-Generator with shared discriminator. **(c)** Shared feature extractor. We omit the input/output names for clarity.

In general, computing the maximum in Eq. 2 is intractable. Therefore, [1] suggested estimating the 1-Wasserstein distance using a pseudo-metric $W_{\mathcal{C}}(\mathcal{D}_1, \mathcal{D}_2)$, where $d$ is parameterized using a neural network from a wide class $\mathcal{C}$. The method minimizes the adversarial loss, derived from Eq. 2,

$$\mathcal{L}_{adv}(g_i, d_i) := \mathop{\mathbb{E}}_{z_{1:i}} \left[ d_i(u_{\hat{I},i}) \right] - \mathop{\mathbb{E}}_{u_{I,i}} \left[ d_i(u_{I,i}) \right], \tag{3}$$

where $z_{1:i} = (z_1, \ldots, z_i)$, $u_{\hat{I},i} \sim \mathcal{D}_{\hat{I},i}$ and $u_{I,i} \sim \mathcal{D}_{I,i}$. The above objective is minimized with respect to the parameters of $g_i$ and maximized with respect to the parameters of the discriminator $d_i$, while freezing the parameters of all previous generators $g_1, \ldots, g_{i-1}$. Note that $\hat{I}_i$ is given by $g_i$ according to Eq. 1. In order to guarantee that $d_i$ is of a bounded Lipschitz constant, in [7] they apply an additional gradient penalty loss to regularize the Lipschitzness of the discriminator:

$$\mathcal{L}_{lip}(d_i) := \mathop{\mathbb{E}}_{u_{I,i}} \left[ \|\nabla_u d_i(u_{I,i})\|_2^2 \right], \tag{4}$$

In addition, they employ a reconstruction loss. We let $z_1^0$ be fixed random noise, such that:

$$\hat{I}_1^0 := \hat{g}_1(z_1^0), \quad \hat{I}_i^0 := \hat{I}_{i-1}^{0,\uparrow} + \hat{g}_i(\hat{I}_{i-1}^{0,\uparrow}), \quad i > 1 \tag{5}$$

In practice, the expected values with respect to the various distributions are replaced with averages over finite sample sets. For simplicity, throughout the paper we will use expectations to avoid clutter.

## 2.2 Hypernetworks

Formally, a hypernetwork $h(z; f(I; \theta_f))$ is a pair of collaborating neural networks, $f$ and $h$. For an input $I$, network $f$, parameterized by a set $\theta_f$ of trainable parameters, produces the weights $\theta_I = f(I; \theta_f)$ for the *primary network* $h$. Network $h$ takes an input $z$ and returns an output $h(z; \theta_I)$ that depends on both $z$ and the task-specific input $I$. In practice, $f$ is typically a large neural network and $h$ is a small neural network. Throughout the paper, we use ";" to separate the input and trainable parameters of a neural network.

## 3 Method

Our method solves an inherent limitation of current single-sample GAN models, which is the scaling to multi-sample learning, such that the same network can perform single-image generations for each sample. For this purpose, we adopt a hypernetwork-based modeling for the generators and discriminators involved. The hypernetwork network $f_g$ produces weights for each $g_i$, and a hypernetwork $f_d$ produces weights for each $d_i$. In this setting, $g_i$ and $d_i$ consist of the same architecture presented in Sec. 2, and serve as the primary networks for $f_g$ and $f_d$ (resp.).

Two alternatives for the proposed setting are presented for completeness and are briefly discussed in Sec. 4, (i) shared discriminator and (ii) shared feature extractor. These alternatives help in understanding our proposed approach. The full description and proofs are presented in the supplementary material. An illustration of the proposed model and the two variants are presented in Fig. 1.

## 3.1 The Model

Our model consists of two main components: a *hyper-generator* and a *hyper-discriminator* (see Fig 1(a) for an illustration). The hyper-generator $g_i$ is a hypernetwork defined as follows:

$$g_i(z, I) := g_i(z; f_g^i(I; \theta_{f_g})),\qquad(6)$$

where $f_g^i(I; \theta_{f_g})$ is a neural network that takes an input image $I$ and returns a vector of weights for the $i$'th generator $g_i$. This network is decomposed into an embedding network $e_g$ that is shared among scales and a linear projection $p_g^i$ per scale,

$$E_g(I) := e_g(I; \theta_{e_g})\qquad(7)$$
$$f_g^i(I; \theta_{f_g}) := p_g^i(E_g(I); \theta_{p_g}^i)\qquad(8)$$

The network $e_g$ is parameterized with a set of parameters $\theta_{e_g}$, and each projection $p_g^i$ is parameterized with a set of parameters $\theta_{p_g}^i$ (for simplicity, we denote their union by $\theta_{p_g} = (\theta_{p_g}^i)_{i=1}^k$). Each $g_i$ is a fully convolutional network, following Sec. 2, whose weights are $\theta_g^i := f_g^i(I; \theta_{f_g})$. The overall set of trainable parameters within $g_i$ is $\theta_{f_g} := (\theta_{e_g}, \theta_{p_g}^i)_{i=1}^k$. The hyper-discriminator is defined in a similar manner:

$$d_i(u, I) := d_i(u; f_d^i(I; \theta_{f_d}))\qquad(9)$$

where $f_d^i(I; \theta_{\theta_d})$ is a network that takes an image $I$ and returns a vector of weights for the $i$'th discriminator $d_i$. This network is also decomposed into an embedding network and a set of projections:

$$E_d(I) := e_d(I; \theta_{e_d})\qquad(10)$$
$$f_d^i(I; \theta_{f_d}) := p_d(E_g(I); \theta_{p_d}^i)\qquad(11)$$

In contrast to the generator, the hyper-discriminator works only on the last image scale. Each $d_i$ is a fully convolutional network, following Sec. 2, whose weights are $\theta_d^i := f_d^i(I; \theta_{f_d})$. The overall set of trainable parameters within $d_i$ is $\theta_{f_d} := (\theta_{e_d}, \theta_{p_d}^i)_{i=1}^k$.

## 3.2 Loss Functions

Our objective function is decomposed into an adversarial and a reconstruction loss function,

$$\mathcal{L}(g_i, d_i) = \mathcal{L}_{adv}(g_i, d_i) + \lambda_1 \cdot \mathcal{L}_{lip}(d_i) + \lambda_2 \cdot \mathcal{L}_{acc-rec}(g_i),\qquad(12)$$

where $\lambda_1, \lambda_2 > 0$ are two tradeoff parameters. The loss functions are described below.

**Adversarial Loss Function** Our adversarial loss function is defined in the following manner:

$$\mathcal{L}_{adv}(g_i, d_i) := \mathbb{E}_I \left\{ \mathop{\mathbb{E}}_{z_{1:i}} d_i(u_{\hat{I},i}; f_d^i(I)) - \mathop{\mathbb{E}}_{u_{I,i}} d_i(u_{I,i}; f_d^i(I)) \right\},\qquad(13)$$

which is maximized by $\theta_{f_d}$ and minimized by $\theta_{f_g}$. In order to suffice that $d_i$ would have a bounded Lipschitz constant, we apply the gradient penalty loss function:

$$\mathcal{L}_{lip}(d_i) := \mathbb{E}_I \mathbb{E}_{u_{I,i}} \left[ \|\nabla_u d_i(u_{I,i})\|_2^2 \right]\qquad(14)$$

For a theoretical analysis of the sufficiency of these loss functions, see Sec. 4.

**Reconstruction Loss Function** Our method also employs a similar loss function to the reconstruction loss defined in Sec. 2. We accumulate all previous reconstruction losses for each scale:

$$\mathcal{L}_{acc-rec}(g_i) := \mathbb{E}_i \mathcal{L}_{rec}(\hat{I}_i, I_i)\qquad(15)$$

Previous methods [31, 13, 8] freeze each intermediate generator $g_i$, except for the current training scale, ensuring that each $g_i$ is independent [2]. In our case, we freeze the projection of all previous scales, except the current scale. However, because $e_g$ is shared for all projections, the accumulated reconstruction loss regularizes the training of $e_g$, by minimizing the reconstruction loss with freezed projections as well. We note that this accumulation is mostly needed for small datasets, whereas for large ones we simply compute the loss with respect to the last scale.

---

[2]In [13] they optimized each $g_i$ with its $j$ (constant) neighboring scales.

### 3.3 Initialization and Optimization

We initialize the hypernetworks with the initialization suggested by [21]. In this initialization, the network $f$ is initialized using the standard Kaiming He initialization [11]. Each convolutional layer in the primary networks $g_i$ and $d_i$ has a $\frac{1}{\sqrt{c_{in} \cdot K \cdot K}}$ normalization, where $c_{in}$ is the number of input channels, and $K \times K$ is the kernel size of the convolution layer.

We train the model progressively from scale $1$ to scale $k$. As noted, during training we freeze all previous projection layers, except for the current training scale. The networks $f_g$ and $f_d$ are continuously trained across scales, where for $f_g$ we add additional projection layers for each new scale, initialized by the previous scale, while $f_d$ does not change. Each scale is trained for a constant number of iterations and optimized using the Adam [17] optimizer. Full training and experiments settings are presented in the supplementary material.

## 4 Theoretical Analysis

In this section, we analyze the soundness of our method, demonstrating its sufficiency. In the supplementary material we demonstrate the importance of the hyper-discriminator and that the generator and discriminator should inherit their parameters from two disjoint hypernetworks. For simplicity, throughout the analysis we omit the reconstruction loss (i.e., $\lambda_2 = 0$), and assume that the distributions $\mathcal{D}_I$, $\mathcal{D}_{\hat{I},i}$ and $\mathcal{D}_{I,i}$ are supported by bounded sets. The proof for each proposition is given in the supplementary material.

We are interested in finding a hyper-generator $g_i$ for each scale, such that, for each image $I$, $g_i(\cdot, I)$ would generate samples $\hat{I}_i$ whose patches $u_{\hat{I},i} \sim \mathcal{D}_{\hat{I},i}$ are similar to the patches $u_{I,i} \sim \mathcal{D}_{I,i}$ within $I_i$. Specifically, we would like to train the parameters of $g_i$ to minimize the following function:

$$\mathbb{E}_I W_\mathcal{C}(\mathcal{D}_{\hat{I},i}, \mathcal{D}_{I,i}) = \mathbb{E}_I \max_{d_i^I \in \mathcal{C}^1} \big\{ \mathop{\mathbb{E}}_{z_{1:i}} d_i^I(u_{\hat{I},i}) - \mathop{\mathbb{E}}_{u_{I,i}} d_i^I(u_{I,i}) \big\}, \tag{16}$$

where $\mathcal{C}^\alpha := \mathcal{C} \cap \{d_i \mid \|d_i\|_L \le \alpha\}$. However, to minimize this objective function directly, one needs to be able to hold a different discriminator $d_i^I$ for each sample $I$, which is computationally expensive.

Fortunately, we can think about this expression in a different manner, as it can also be written as follows:

$$\mathbb{E}_I W_\mathcal{C}(\mathcal{D}_{\hat{I},i}, \mathcal{D}_{I,i}) = \max_S \mathbb{E}_I \big\{ \mathop{\mathbb{E}}_{z_{1:i}} d_i(u_{\hat{I},i}; S(I)) - \mathop{\mathbb{E}}_{u_{I,i}} d_i(u_{I,i}; S(I)) \big\}, \tag{17}$$

where the maximum is taken over the set of mappings $S$ from images $I$ to parameters $\theta_I$ of discriminators $d_i^I \in \mathcal{C}^1$. We let $S^*$ be a mapping that takes $I$ and returns the parameters $S^*(I)$ of the discriminator $d_i^I := d_i(\cdot; S^*(I)) = \arg\max_{d_i \in \mathcal{C}^1} \big\{ \mathbb{E}_{z_{1:i}} d_i^I(u_{\hat{I},i}) - \mathbb{E}_{u_{I,i}} d_i^I(u_{I,i}) \big\}$.

Therefore, if $S^*$ can be approximated by a large neural network $f_d^i(I) = f_d^i(I; \theta_{f_d}) \approx S^*(I)$, then we can approximately solve the maximization in Eq. 17 by parameterizing the discriminator with a hypernetwork $d_i := d_i(u; f_d(I; \theta_{f_d}))$ and training its parameters to solve the maximization in Eq. 17 (approximately). For instance, if $S^*$ is a continuous function, one can approximate $S^*$ using a large enough neural network up to any approximation error $\le \epsilon$ [4, 14, 27, 24, 10, 20, 30]. This is summarized in the following proposition.

**Proposition 1.** *Assume that $\mathcal{I} \subset \mathbb{R}^{3 \times h \times w}$ is compact. Let $\epsilon > 0$ be an approximation error. Let $g_i(z, I) := g_i(z; f_g^i(I; \theta_{f_g}))$ be a hyper-generator and $\mathcal{C}$ a class of discriminators. Assume that $S^*$ is continuous over $\mathcal{I}$. Then, there is a large enough neural network $f_d^i$ (whose size depends on $\epsilon$), such that, the hyper-discriminator $d_i(u, I) := d_i(u; f_d^i(I; \theta_{f_d}))$ satisfies:*

$$\mathbb{E}_I W_\mathcal{C}(\mathcal{D}_{\hat{I},i}, \mathcal{D}_{I,i}) = \max_{\theta_{f_d}} \mathcal{L}_{adv}(g_i, d_i) + o_\epsilon(1), \tag{18}$$

*where the maximum is taken over the parameterizations $\theta_{f_d}$ of $f_d$, such that, $d_i(\cdot; f_d^i(I; \theta_{f_d})) \in \mathcal{C}^1$.*

A proof for the existence of a continuous selector $S^*(I)$ has been proposed [6, 25, 26] for similar settings, and the proof for Prop. 1 is provided as part of the supplementary material. According to this proposition, in order to minimize $\mathbb{E}_I W_\mathcal{C}(\mathcal{D}_{\hat{I},i}, \mathcal{D}_{I,i})$, we can simply parameterize

our discriminator with a hypernetwork $d_i := d_i(u; f_d^i(I; \theta_{f_d}))$ and train the hyper-generator $g_i$ to solve: $\min_{\theta_{f_g}} \max_{\theta_{f_d}} \mathcal{L}_{adv}(d_i, g_i)$ along with the gradient penalty loss $\mathcal{L}_{lip}(d_i)$ to ensure that $d_i(\cdot; f_d(I; \theta_{f_d}))$ would have a bounded Lipschitz constant.

Put differently, in order to guarantee that the approximation error in Prop. 1 would be small, we advocate **selecting the hypernetwork $f_d^i$ to be a large neural network**. In this case, if we are able to effectively optimize $\theta_{f_g}$ and $\theta_{f_d}$ to solve $\min_{\theta_{f_g}} \max_{\theta_{f_d}} \mathcal{L}_{adv}(d_i, g_i)$ (s.t the Lipschitz constant of $d_i$ is bounded), we can ensure that $\mathbb{E}_I W_\mathcal{C}(\mathcal{D}_{\hat{I},i}, \mathcal{D}_{I,i})$ is small, as desired.

### 4.1 Alternative Architectures

As presented in Sec. 3, we consider two alternative architectures (i) shared discriminator and (ii) shared feature extractor. We briefly describe each proposed variant and its limitations; the full analysis is presented in the supplementary material.

**Shared Discriminator**   In this case, the model has two main components for each scale $i$: a hyper-generator $g_i(z, I) = g_i(z; f_g^i(I; \theta_{f_g}))$, along with a standard discriminator $d_i(u) = d_i(u; \theta_d)$ that is shared among all samples $I$, as illustrated in Fig. 1(b). We show that if the expected (w.r.t $I \sim \mathcal{D}_I$) distance between the distributions $\mathcal{D}_{\hat{I},i}$ and $\mathcal{D}_i$ is small, then the loss function $\mathcal{L}_{adv}(g_i, d_i) := \mathbb{E}_I\{\mathbb{E}_{z_{1:i}} d_i(u_{\hat{I},i}) - \mathbb{E}_{u_{i,I}} d_i(u_{I,i})\}$ tends to be small. Here, $\mathcal{D}_i$ denotes the distribution of $u_{\hat{I},i} \sim \mathcal{D}_{\hat{I},i}$ for $I \sim \mathcal{D}_I$. This proposition shows that a hyper-generator $g_i(\cdot, I)$ that generates samples $\hat{I}_i$ whose patches are similar to samples of $\mathcal{D}_i$ would minimize the loss function $\mathcal{L}_{adv}(g_i, d_i)$, even though the generated samples are not conditioned on the image $I$. Therefore, simply minimizing the adversarial loss would not guarantee that $g_i(\cdot, I)$ generates samples $\hat{I}_i$ that are similar to $I_i$.

**Shared Feature Extractor**   We note that as a strategy for reducing the number of trainable parameters in the whole model, one could restrict $f_g$ and $f_d$ to share their encoding component $e$, as illustrated in Fig. 1(c). We show two cases in which this approach fails. First, we consider the case where the model is trained using GD. In this case, GD iteratively updates $(\theta_e, \theta_{p_g}^i)$ to minimize $\mathcal{L}_{adv}(g_i, d_i)$ and updates $(\theta_e, \theta_{p_d}^i)$ to maximize $\mathcal{L}_{adv}(g_i, d_i) - \lambda_1 \cdot \mathcal{L}_{lip}(d_i)$. Informally, we show that $\theta_e$ is essentially trained to only minimize $\mathcal{L}_{lip}(d_i)$ and that each tuple $(\theta_e, \theta_{p_g}^i, \theta_{p_d}^i)$ with $d_i \equiv 0$ is an equilibrium point. In addition, we note that $\mathcal{L}_{lip}(d_i)$ is minimized by $d_i \equiv 0$. Therefore, it is likely that $d_i$ would converge to 0 during training. This means that at some point the discriminator is ineffective. In particular, if $\theta_e = 0$, then $(\theta_e, \theta_{p_g}^i, \theta_{p_d}^i)$ is an equilibrium point. We note that $\theta_e = 0$ is not a desirable output of the training algorithm, since it provides a hyper-generator $g_i(\cdot, I)$ that is independent of the input image $I$. Second, we consider the case where GD iteratively optimizes $(\theta_e, \theta_{p_g}^i)$ to minimize $\mathcal{L}_{adv}(g_i, d_i)$, $\theta_{p_d}^i$ to maximize $\mathcal{L}_{adv}(g_i, d_i)$ and $(\theta_e, \theta_{p_g}^i)$ to minimize the loss $\mathcal{L}_{lip}(d_i)$. We show that each tuple $(\theta_e, \theta_{p_g}^i, \theta_{p_d}^i)$ with $\theta_e = 0$ is, again, an equilibrium point.

## 5 Experiments

Our experiments are divided into two parts. In the first part, we study three different training regimes of our method. First, we experiment with single-image training, in order to produce a fair comparison with existing methods. Second, we present a mini-batch training scheme, where instead of a single image, the model is trained on a fixed set of images. Lastly, we experiment with training over a full dataset that cannot fit into a single batch.

In the second part, we experiment with several applications of our method. Specifically, we study the ability of our method in the Harmonization, Editing and Animation tasks proposed by [31], as well as generating samples of arbitrary size and aspect ratio. In addition, we experiment with two new applications: image interpolations, and generation at inference time. These application are unique to multi-image training.

Due to space constraints, we focus on our novel applications, and refer the reader to the supplementary material for our full set of applications, as well as technical details, such as specific hyperparameters, GPU usage and additional experiments.

Throughout the experiments, we consider the following set of baselines: SinGAN [31], ConSin-GAN [13] and HP-VAE-GAN [8]. To evaluate image generation, we use the single-image FID metric

Table 1: **Quantitative comparison on Places-50/LSUN-50**, showing SIFID, mSIFID, diversity and training time per image (minutes). Our method shows comparable results to single-image models in both *single* and *dataset* settings, where the overall training time per image is significantly lower.

| Method | SIFID ↓ | mSIFID ↓ | Diversity↑ | min./image↓ |
|---|---|---|---|---|
| SinGAN [31] | 0.09/0.11 | 0.15/0.20 | 0.52/0.60 | 60 |
| ConSinGAN [13] | 0.06/0.08 | 0.08/0.13 | 0.50/0.55 | 20 |
| HP-VAE-GAN [8] | 0.17/0.40 | 0.27/0.62 | 0.62/0.78 | 60 |
| Ours *Single* | 0.03/0.11 | 0.06/0.19 | 0.57/0.65 | 30 |
| Ours *Dataset* | 0.05/0.11 | 0.07/0.16 | 0.50/0.48 | 5 |

Table 2: **Varying the batch size in single mini-batch training.** Both SIFID and diversity (w.r.t a specific batch size) remain stable regardless of the size of the mini-batch.

| Batch Size | SIFID↓ | mSIFID↓ | Diversity↑ |
|---|---|---|---|
| 1 | 0.03 | 0.07 | 0.73 |
| 2 | 0.04 | 0.07 | 0.66 |
| 3 | 0.03 | 0.07 | 0.68 |
| 4 | 0.04 | 0.08 | 0.70 |
| 5 | 0.04 | 0.08 | 0.71 |

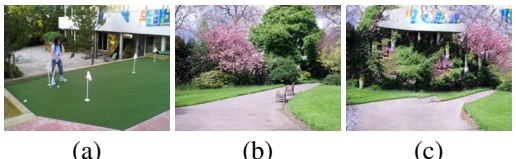

(a)      (b)      (c)

Figure 2: **Leakage in the multi-image training when using a shared discriminator. (a)** and **(b)** are the two training images, and **(c)** is a generated image for the model of image **(b)**. As can be seen, it contains patches from the first image as well.

Table 3: **Performance on the Valley dataset**, showing SIFID, mSIFID, diversity and training time per image (minutes). As can be seen, inference performance increases with training data size.

| Dataset | Train | | | | Test | | |
|---|---|---|---|---|---|---|---|
| | SIFID↓ | mSIFID↓ | Diversity↑ | min./image↓ | SIFID↓ | mSIFID↓ | Diversity↑ |
| Valley$_{500}$ | 0.04 | 0.07 | 0.51 | 4.0 | 0.47 | 2.47 | 0.34 |
| Valley$_{2500}$ | 0.05 | 0.08 | 0.52 | 3.5 | 0.43 | 1.86 | 0.37 |
| Valley$_{5000}$ | 0.05 | 0.08 | 0.51 | 3.0 | 0.41 | 1.52 | 0.40 |

(SIFID) [31]. Following [31], the metric represents the mean of minimum SIFID over 100 generated samples per image. We further compute the mean-SIFID (mSIFID), which is the mean across all generated samples of all images, without taking the minimum for each image.

As simply overfitting the training image would lead to a SIFID value of 0, a diversity measure is necessary. For this purpose we employ the diversity measure used in [31]. This measure is computed as the average standard deviation over all pixel values along the channel axis of 150 generated images.

Previous works in the field [32, 31] require training on each image independently. To enable comparison with previous work, we use the 50-image dataset of [31], denoted by Places-50 and the 50-image dataset of [12], denoted by LSUN-50. Additionally, whenever a quantitative measure is available, we present competitive results, and qualitatively our results are at least as good, if not better than those of single-image GANs. For larger datasets of up to 5000 images we perform comprehensive experiments with our proposed method.

The dataset presented by SinGAN, Places-50, consists of 50 images selected at random from sub-categories of the Places dataset [38] – Mountains, Hills, Desert and Sky and the dataset presented by ConSinGAN, LSUN-50, consists of five randomly sampled images from each of the ten LSUN dataset categories. In order to evaluate our method on larger datasets, we consider three subsets of the *Valleys* category of the Places dataset; the first 500(V500), 2500(V2500) and 5000(V5000) (the entire category) images, and use the 100-image test set when relevant. Additionally, we consider the first 250(C250) and 500(C500) images of the Churches Outdoor category of the LSUN dataset.

## 5.1 Training Procedures

**Single-Image training**    Our approach is first evaluated when training with a single image, as done in previous methods. Since a single function needs to be learned, a standard discriminator (*i.e.*, not learned via hypernetwork) is used in this specific case in order to avoid redundant enlargement of the model and speed up training. Similar results are obtained using a hyper-discriminator. Tab. 1 shows that our performance is on par with current single-image models for this setting.

**Single mini-batch training**    When introduced with multiples images, the standard discriminator, as for the baseline methods, suffers from leakage between the images in the mini-batch, *i.e.*, the patches of the generated images are distributed as patches of arbitrary images from the batch (Sec. 4.1– Shared Discriminator). Fig. 2 illustrates this effect. To overcome this issue, we introduce a hyper-discriminator that enables efficient learning of a different discriminator model for each image. To evaluate performance on single mini-batch learning, we randomly sampled a set of 5 images from the 50-image dataset and trained a different model for each permutation of the set of size $1 \leq i \leq 5$. Tab. 2 show performance is good regardless of the mini-batch size, which indicates that the hypernetwork model successfully learns a different model for each input image.

**Dataset training**    Our main contribution arises from training with large amount of data. In addition to Places-50 and LSUN-50, we trained our method on three subsets of the Valley category, as presented above – $\text{Valley}_{500}$, $\text{Valley}_{2500}$ and $\text{Valley}_{5000}$, iterating on batches of 16 images, for 25k, 100k, 150k iterations per scale, respectively, on a single GPU. Tab. 1 and 3 show performance and training time per image. $\text{Churches}_{250}$ and $\text{Churches}_{500}$ were trained in a similar manner for 20k and 30k iterations per scale, respectively, and reached equal performance of 0.20, 0.27 and 0.47 for SIFID, mSIFID and diversity metrics.

As far as we are aware, our method is the first that is able to train multiple single-image models at this scale with a decent runtime. For example, training the model presented by [31] on $\text{Valley}_{5000}$ with a single GPU would require the training of 5000 different and independent models, and would take approximately 200 days. Our method takes 10 days on a single GPU, and thus is faster by a factor of 20.

## 5.2 Applications

As noted above, we present our novel applications in the main text, and refer the reader to the supplementary material for applications presented in previous works.

**Interpolation**    As our meta-learning approach learns the space of generators, and is trained on multiple images, our model is able to interpolate between different images smoothly and at different scales. Unlike common interpolation, the hierarchical structure of the model enables interpolation at different scales as follows: We start by interpolating over the latent representation $e_g$, resulting in a new *generator*. Let $A$ and $B$ be two different images. We compute their latent representations $e_g^A = E_g(A)$ and $e_g^B = E_g(B)$ (resp.) and perform linear interpolation between the two, for $\alpha \in [0,1]$, $e_g^\alpha = \alpha e_g^A + (1-\alpha)e_g^B$, resulting in a new generator. We then select a primary image, $A$ for example, and initial scale $m$, and perform the following steps: (i) we use $e_g^A$ to generate scales 1 to $m$, and (ii) from scale $m$, we switch to $e_g^\alpha$, and continue generating scales accordingly. The result is a mix at different patch scales, where scale 1 controls the structure of the image, and the last scale controls the finer texture of the image. Fig. 3 shows an example of a pair of images and its interpolation, where the primary image is denoted by $A$ (top-left) and the target image by $B$ (top-right). We show interpolation at three different scales - first (1), middle, and last, presented from top to bottom. As can be seen, interpolating at the first scale results in more structural changes, while interpolating at the middle and last scales results in more textural changes. By changing $\alpha$ we are able to obtain a wide gamut of intermediate options between the two images.

**Feedforward generation**    The meta-learning approach, and the fact that our method is able to learn from a relatively large dataset such as $\text{Valley}_{5000}$, introduce the ability to model a new image in one forward pass. Fig. 5 and Tab. 3 show inference results for three different models trained on $\text{Valley}_{500}$, $\text{Valley}_{2500}$ and $\text{Valley}_{5000}$. As can be seen, it requires a significantly larger dataset than that of [31] to get the model to generalize. The network trained on $\text{Valley}_{5000}$ enables modeling of a new image in a fraction of a second, and results in coherent and reasonable generated images compared with previous methods, which were unable to perform this task.

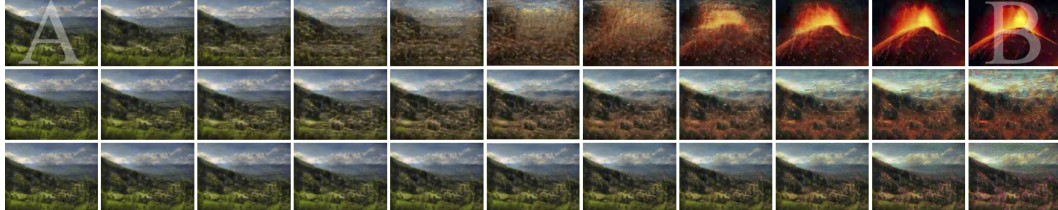

Figure 3: **Interpolation in the space of generative networks**. A hypernetwork is trained to produce unique Single-Image Generators from a dataset of 50 images. **Top left (right)** - a generated image from generator A (B). Each column represents different mixtures of the generators' latent representations. Each row represents injection of the mixed representation at different scales, where all previous scales use generator A representation - from coarsest **(top)** to finest **(bottom)**.

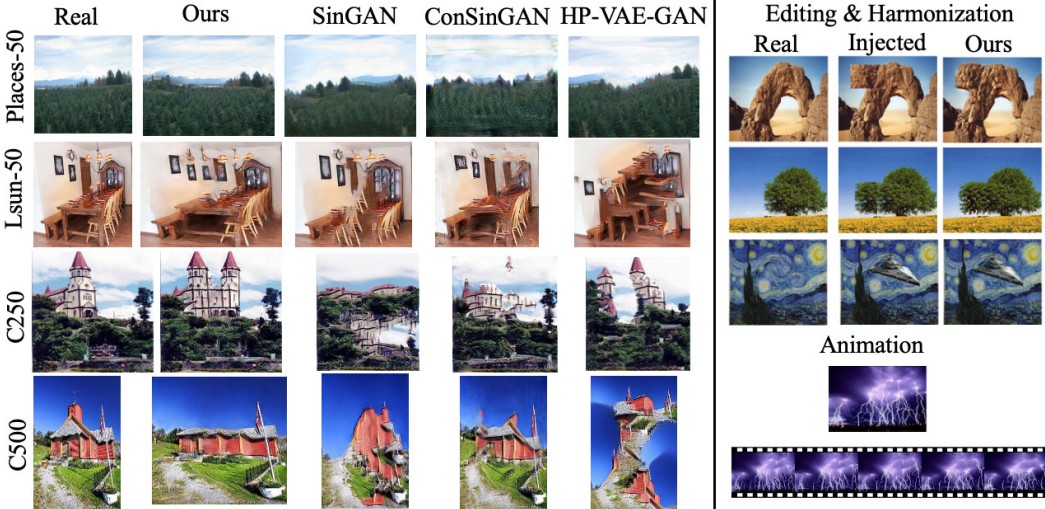

Figure 4: **Left:** Comparison of image generation results with single image baselines, on different datasets. **Right:** Results of applications, trained with the Places-50 dataset. Our method allows us to manipulate images such as Editing, Harmonization and Animation at a large scale, training all images at once.

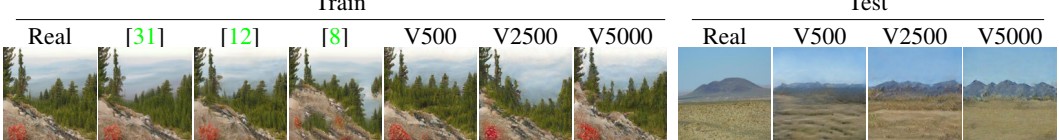

Figure 5: **Training and testing results on Valley dataset**. Real images from the train/test-set respectively. Training results include SinGAN [31], ConSinGAN [12] and HP-VAE-GAN [8].

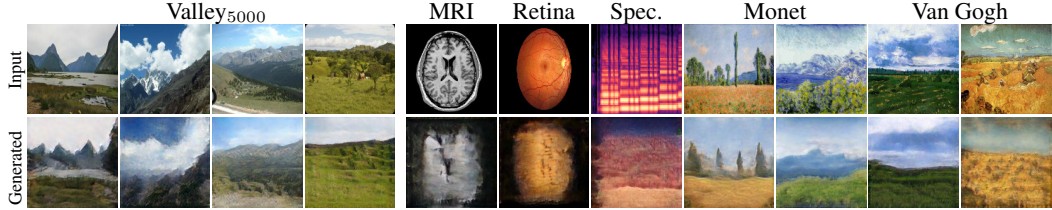

Figure 6: **Feedforward generation** with the Valley$_{5000}$ trained model on unseen images: **Left-side** - image form the same distribution. **Right-side** - images from completely different distributions.

# 6 Limitations

A prominent limitation of the method is the model's size. The hypernetwork approach leads to large projection layers, *e.g.* a convolution layer with a weight size of $(64, 64, 3, 3)$ and embedding size of $|e| = 512$ will result in a projection layer with a weight size of $|(512, 36864)| \approx 18M$ parameters. This obviously affects convergence, runtime and GPU memory usage.

In Fig. 6, we quantitatively explore the out-of-distribution generalization capabilities of our feedforward method when training on the Valley$_{5000}$ nature image dataset. As can be seen, for images that are completely out of domain, the generated images are not faithful to the input image. Training on a large-scale heterogeneous dataset to further improve generalization requires days of training. Until this experiment is performed, it is unclear whether the architecture has enough capacity to support this one-model-fits-all capability.

# 7 Related work

Hypernetworks, which were first introduced under this name in [9], are networks that generate the weights of a second *primary* network, which computes the actual task. Hypernetworks are especially suited for meta-learning tasks, such as few-shot [2] and continual learning tasks [36], due to the knowledge-sharing ability of the weight-generating network. Knowledge sharing in hypernetworks was recently used for continual learning by [36].

Predicting the weights instead of performing backpropagation can lead to efficient neural architecture search [3, 37] and hyperparameter selection [23]. In [22], hypernetworks were applied for 3D shape reconstruction from a single image. In [34] hypernetworks were shown to be useful for learning shared image representations. Note that while the name of the method introduced in [28] is reminiscent of our method, it solves a different task, with a completely different algorithm. Their method does not employ a hypernetwork to parameterize their generator (or discriminator); instead, their generator serves as a hypernetwork itself. In addition, they intend to learn the distribution of weights of high-performing classifiers on a given classification task, which is a different application.

Several GAN-based approaches were proposed for learning from a single image sample. Deep Image Prior [35] and Deep Internal Learning [33] showed that a deep convolutional network can form a useful prior for a single image in the context of denoising, super-resolution, and inpainting. SinGAN [29] uses patch-GAN [15, 29, 18, 19] to model the multiscale internal patch distribution of a single image, thus generating novel samples. ConSinGAN [12] extends SinGAN, improving quality and training time. However, these methods need to be trained on each image individually. In this work, we propose a novel approach based on hypernetworks that leverages the capabilities of single-image generation and enables efficient training on datasets of any size, while preserving the unique properties of single-image training.

# 8 Conclusions

Given the abundance of unlabeled training images, training a single image GAN is unjustifiable if viable multi-image alternatives exist. We present the first such alternative, which also opens the door to novel applications that are impossible with existing models, such as interpolation between single-image domains and feedforward modeling. From a technical perspective, we present what is to our knowledge the first adversarial hypernetwork. Working with this novel multi-network structure requires an understanding of the interplay between the components involved, and we support our method with theoretical analysis.

## Acknowledgments

This project has received funding from the European Research Council (ERC) under the European Union's Horizon 2020 research and innovation programme (grant ERC CoG 725974). The contribution of the first author is part of a Master thesis research conducted at Tel Aviv University.

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
