# Meta Internal Learning : Supplementary material

**Raphael Bensadoun**[*]
The School of Computer Science
Tel Aviv University

**Shir Gur**
The School of Computer Science
Tel Aviv University

**Tomer Galanti**
The School of Computer Science
Tel Aviv University

**Lior Wolf**
The School of Computer Science
Tel Aviv University

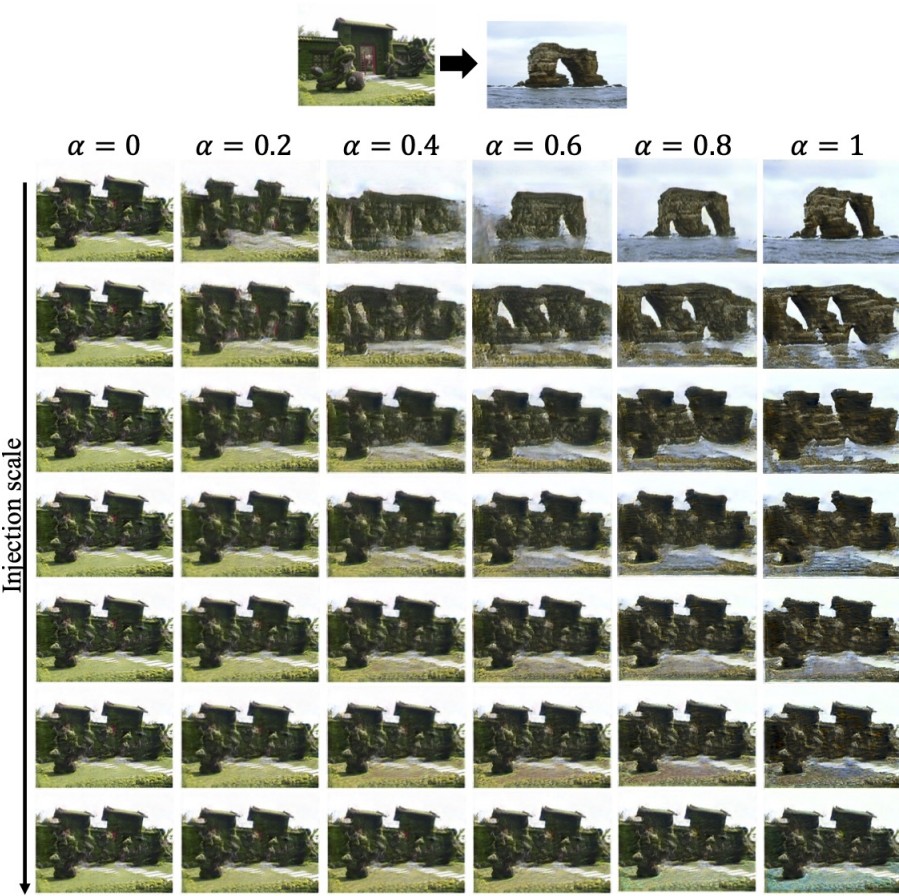

Figure 1: **Interpolation in the space of generative networks:** Interpolating at different scales and different interpolation coefficients creates a wide gamut of intermediate options between the two generated images.

---
[*]Corresponding author - bensadoun@mail.tau.ac.il

35th Conference on Neural Information Processing Systems (NeurIPS 2021).

# Contents

# 1 Theoretical Analysis

## 1.1 Shared Discriminator

In this section, we consider the case where the model has two main components: a hyper-generator $g_i(z, I) = g_i(z; f_g^i(I; \theta_{f_g}))$ along with a standard discriminator $d_i(u) = d_i(u; \theta_d)$ that is shared among all samples $I$. In this setting, the adversarial objective function is defined as:

$$\mathcal{L}_{adv}(g_i, d_i) := \mathbb{E}_I \big\{ \underset{z_{1:i}}{\mathbb{E}} \, d_i(u_{\hat{I}, i}) - \underset{u_{i,I}}{\mathbb{E}} \, d_i(u_{I,i}) \big\} \tag{1}$$

The following proposition shows that if the expected (with respect to the distribution of $I$) distance between the distributions $\mathcal{D}_{\hat{I},i}$ and $\mathcal{D}_i$ is small, then, the loss function $\mathcal{L}_{adv}(g_i, d_i)$ tends to be small. For this purpose, we assume that $\mathcal{C}$ is closed under multiplication by positive scalars (i.e., $\alpha d \in \mathcal{C}$ for all $d \in \mathcal{C}$ and $\alpha > 0$). This is a technical assumption that holds for any set of neural networks, with a linear top-layer.

**Proposition 1.** *Let $g_i(z, I) := g_i(z; f_g^i(I; \theta_{f_g}))$ be a hyper-generator and $d_i \in \mathcal{C}$ a shared discriminator at scale $i$. Let $\mathcal{D}_i$ be the distribution of $u \sim \mathcal{D}_{I,i}$, where $I \sim \mathcal{D}_I$. Assume that $\mathcal{C}$ is closed under multiplication by positive scalars. Then,*

$$\mathcal{L}_{adv}(g_i, d_i) \leq \|d_i\|_L \cdot \mathbb{E}_I[W_{\mathcal{C}^1}(\mathcal{D}_{\hat{I},i}, \mathcal{D}_i)] \tag{2}$$

*In particular, $\max_{d_i \in \mathcal{C}^1} \mathcal{L}_{adv}(g_i, d_i) \leq \mathbb{E}_I[W_{\mathcal{C}}(\mathcal{D}_{\hat{I},i}, \mathcal{D}_i)]$.*

*Proof.* Let $\alpha := \|d_i\|_L$, $u_{I,i} \sim \mathcal{D}_{I,i}$ (conditioned on a fixed $I$) and let $u \sim \mathcal{D}_i$ be a random variable. We can write:

$$\mathcal{L}(g_i, d_i) \leq \max_{d_i' \in \mathcal{C}^\alpha} \mathcal{L}(g_i, d_i')$$

$$= \max_{d_i' \in \mathcal{C}^\alpha} \underset{I}{\mathbb{E}} \left\{ \underset{z_{1:i}}{\mathbb{E}} \, d_i'(u_{\hat{I},i}) - \underset{u_{I,i}}{\mathbb{E}} \, d_i'(u_{I,i}) \right\}$$

$$= \max_{d_i' \in \mathcal{C}^\alpha} \left\{ \underset{I_1}{\mathbb{E}} \underset{z_{1:i}}{\mathbb{E}} \, d_i'(u_{\hat{I},i}) - \underset{I_2}{\mathbb{E}} \underset{u_{I,i}}{\mathbb{E}} \, d_i'(u_{I,i}) \right\}$$

$$= \max_{d_i' \in \mathcal{C}^\alpha} \left\{ \underset{I_1}{\mathbb{E}} \underset{z_{1:i}}{\mathbb{E}} \, d_i'(u_{\hat{I},i}) - \underset{u}{\mathbb{E}} \, d_i'(u) \right\}$$

$$= \max_{d_i' \in \mathcal{C}^\alpha} \underset{I}{\mathbb{E}} \left\{ \underset{z_{1:i}}{\mathbb{E}} \, d_i'(u_{\hat{I},i}) - \underset{u}{\mathbb{E}} \, d_i'(u) \right\},$$

where $I_1, I_2 \sim \mathcal{D}_I$ are two i.i.d. random variables. We note that for any real-valued function $f$, we have: $\max_x \mathbb{E}_y[f(x, y)] \leq \mathbb{E}_y[\max_x f(x, y)]$. Therefore,

$$\mathcal{L}(g_i, d_i) \leq \underset{I}{\mathbb{E}} \max_{d_i' \in \mathcal{C}^\alpha} \left\{ \underset{z_{1:i}}{\mathbb{E}} \, d_i'(u_{\hat{I},i}) - \underset{u}{\mathbb{E}} \, d_i'(u) \right\} \tag{3}$$

We note that any function $d \in \mathcal{C}^1$ can be translated into a function $\alpha d \in \mathcal{C}^\alpha$ and vice versa since $\mathcal{C} = \alpha \cdot \mathcal{C}$. In particular, $\mathcal{C}^\alpha = \alpha \cdot \mathcal{C}^1$. Hence, we have:

$$\mathcal{L}(g_i, d_i) \leq \underset{I}{\mathbb{E}} \max_{d_i' \in \mathcal{C}^1} \left\{ \underset{z_{1:i}}{\mathbb{E}} \, \alpha \cdot d_i'(u_{\hat{I},i}) - \underset{u}{\mathbb{E}} \, \alpha \cdot d_i'(u) \right\}$$
$$= \alpha \underset{I}{\mathbb{E}}[W_{\mathcal{C}^1}(\mathcal{D}_{\hat{I},i}, \mathcal{D}_i)], \tag{4}$$

which proves the claim. $\square$

This proposition shows that a hyper-generator $g_i(\cdot, I)$ that generates samples $\hat{I}_i$ whose patches are similar to $\mathcal{D}_i$ would minimize the loss function $\mathcal{L}_{adv}(g_i, d_i)$, even though the generated samples are not conditioned on the image $I$. Therefore, minimizing the adversarial loss with a shared discriminator does not guarantee that $g_i(\cdot, I)$ would generate samples $\hat{I}_i$ that are similar to $I_i$, which is undesirable.

## 1.2 Shared Feature Extractor

We note that as a strategy one could reduce the number of trainable parameters in the whole model, by restricting $f_g$ and $f_d$ to share their encoding component $e$, as illustrated in Fig. **??**. In this section, we show two failing cases of this approach. First, we consider the case where $\theta_e$ is optimized to minimize the objectives of both $g$ and $d$. As a second case, we consider the case where $\theta_e$ is optimized to minimize the objective of $g$.

**Case 1** We first consider the case where the model is trained using GD, when $f_g$ and $f_d$ share their representation function's weights. Specifically, GD iteratively updates $(\theta_e, \theta_{p_g}^i)$ to minimize $\mathcal{L}_{adv}(g_i, d_i)$ and updates $(\theta_e, \theta_{p_d}^i)$ to maximize $\mathcal{L}_{adv}(g_i, d_i) - \lambda_1 \cdot \mathcal{L}_{lip}(d_i)$. We denote this optimization process by $\mathbb{A}$. The following proposition shows that $\theta_e$ is trained to minimize $\mathcal{L}_{lip}(d_i)$ only and that $\mathbb{A}$ suffers from a wide span of undesirable equilibrium points.

**Remark 1.** *Let $g_i(z, I) := g_i(z; f_g^i(I; \theta_{f_g}))$ and $d_i(u, I) := d_i(u; f_d^i(I; \theta_{f_d}))$ be the hyper-generator and the hyper-discriminator, with an activation function $\sigma$ that satisfies $\sigma(0) = 0$. Assume that $\theta_{e_g} = \theta_{e_d}$ is shared among $f_g^i$ and $f_d^i$. Then, $\mathbb{A}$ trains $e_g = e_d$ to minimize $\mathcal{L}_{lip}(d_i)$ only. In addition, let $(\theta_e, \theta_{p_g}^i, \theta_{p_d}^i)$ be a set of parameters with $E_g = E_d \equiv 0$. Then, $(\theta_e, \theta_{p_g}^i, \theta_{p_d}^i)$ is an equilibrium point of $\mathbb{A}$.*

*Proof.* We denote $e = e_g = e_d$. Let $\theta_e$, $\theta_{p_g}^i$ and $\theta_{p_d}^i$ be the parameters of $e$, $p_g$ and $p_d^i$. Each iteration of GD updates the weights $(\theta_e, \theta_{p_d}^i)$ of $d_i$ with the following step: $-\mu \frac{\partial \mathcal{L}_{adv}(g_i, d_i)}{\partial(\theta_e, \theta_{p_d}^i)} + \mu \frac{\partial \mathcal{L}_{lip}(d_i)}{\partial(\theta_e, \theta_{p_d}^i)}$. On the other hand, the GD step for the weights $(\theta_e, \theta_{p_g}^i)$ of $g_i$ is $+\mu \frac{\partial \mathcal{L}_{adv}(g_i, d_i)}{\partial(\theta_e, \theta_{p_d}^i)}$. Therefore, since $d_i$ and $g_i$ share weights within their representation function $e$, its update is the sum of the two steps $-\mu \frac{\partial \mathcal{L}_{adv}(g_i, d_i)}{\partial \theta_e}$ and $+\lambda_1 \mu \frac{\partial \mathcal{L}_{adv}(g_i, d_i)}{\partial \theta_e}$ and $-\mu \frac{\partial \mathcal{L}_{lip}(d_i)}{\partial \theta_e}$, which is simply $-\mu \frac{\partial \mathcal{L}_{lip}(d_i)}{\partial \theta_e}$. Therefore, $e$ is trained to minimize $\mathcal{L}_{lip}(d_i)$ using GD.

To see why $(\theta_e, \theta_{p_g}^i, \theta_{p_d}^i)$ (with $E_g \equiv 0$) is an equilibrium point, we notice that $d_i \equiv 0$ is a global minima of $\mathcal{L}_{lip}(d_i)$. In particular, $\theta_e$ would not change when applying $\mathbb{A}$. In addition, we note that if $E_g(I) = E_d(I) = 0$, then, the outputs of $E_g$, $E_d$, $f_g^i$, $f_d^i$, $g_i$ and $d_i$ are all zero, regardless of the values of the weights $\theta_{p_g}^i, \theta_{p_d}^i$, because $\sigma(0) = 0$. Therefore, the gradients of $\mathcal{L}_{adv}(g_i, d_i)$ with respect to $\theta_{p_g}^i$ and $\theta_{p_d}^i$ are zero, and we conclude that $\theta_{p_g}^i$ and $\theta_{p_d}^i$ would not update as well. $\qquad\square$

**Case 2** As an additional investigation, we consider the case where GD iteratively optimizes $(\theta_e, \theta_{p_g}^i)$ to minimize $\mathcal{L}_{adv}(g_i, d_i)$, $\theta_{p_d}^i$ to maximize $\mathcal{L}_{adv}(g_i, d_i)$ and $(\theta_e, \theta_{p_g}^i)$ to minimize the loss $\mathcal{L}_{lip}(d_i)$. We denote this optimization process by $\mathbb{B}$. The following proposition shows that this procedure suffers from a wide span of undesirable equilibrium points.

**Remark 2.** *Let $g_i(z, I) := g_i(z; f_g^i(I; \theta_{f_g}))$ and $d_i(u, I) := d_i(u; f_d^i(I; \theta_{f_d}))$ be a hyper-generator and a hyper-discriminator, both with activation functions $\sigma$ that satisfy $\sigma(0) = 0$. Then, any set of parameters $(\theta_e = 0, \theta_{p_g}^i, \theta_{p_d}^i)$ is an equilibrium point of $\mathbb{B}$.*

*Proof.* We note that if $\theta_e = 0$, then, since $\sigma(0) = 0$, the outputs of $e$, $f_g^i$, $f_d^i$, $g_i$ and $d_i$ are all zero, regardless of the values of the weights $\theta_{p_g}^i, \theta_{p_d}^i$. In particular, the gradients of $\mathcal{L}_{adv}(g_i, d_i)$ with respect to $\theta_{p_g}^i$ and $\theta_{p_d}^i$ are zero. In addition, the Lipschitz loss function is at its global minima for $d_i$, and therefore, its gradient with respect to $(\theta_e, \theta_{p_d}^i)$ is zero as well. Therefore, we conclude that any possible step starting from $(\theta_e = 0, \theta_{p_g}^i, \theta_{p_d}^i)$ would not change the weights. $\qquad\square$

## 1.3 Our Method

**Proposition 2.** *Assume that $\mathcal{I} \subset \mathbb{R}^{3 \times h \times w}$ is compact. Let $\epsilon > 0$ be an approximation error. Let $g_i(z, I) := g_i(z; f_g^i(I; \theta_{f_g}))$ be a hyper-generator and $\mathcal{C}$ a class of discriminators. Assume that $S^*$ is continuous over $\mathcal{I}$. Then, there is a large enough neural network $f_d^i$ (whose size depends on $\epsilon$), such*

*that, the hyper-discriminator $d_i(u, I) := d_i(u; f_d^i(I; \theta_{f_d}))$ satisfies:*

$$\mathbb{E}_I W_{\mathcal{C}}(\mathcal{D}_{\hat{I},i}, \mathcal{D}_{I,i}) = \max_{\theta_{f_d}} \mathbb{E}_I \left\{ \mathbb{E}_{z_{1:i}} d_i(u_{\hat{I},i}; f_d^i(I)) - \mathbb{E}_{u_{I,i}} d_i(u_{I,i}; f_d^i(I)) \right\} + o_\epsilon(1),$$

*where the maximum is taken over the parameterizations $\theta_{f_d}$ of $f_d$, such that, $d_i(\cdot; f_d^i(I; \theta_{f_d})) \in \mathcal{C}^1$.*

*Proof.* Let $\mathcal{S}^1$ be the set of functions $S : I \mapsto \theta_I$, where $\theta_I$ correspond to a discriminator $d_i(\cdot; \theta_I) \in \mathcal{C}^1$. Let $\mathcal{Q}$ be the set of parameters $\theta_{f_d}$, such that, $d_i(\cdot; f_d^i(I; \theta_{f_d})) \in \mathcal{C}^1$ for all $I \in \mathcal{I}$. In particular, for any $\theta_{f_d} \in \mathcal{Q}$, we have: $f_d^i(I; \theta_{f_d})) \in \mathcal{S}^1$. Hence, we have:

$$\mathbb{E}_I W_{\mathcal{C}}(\mathcal{D}_{\hat{I},i}, \mathcal{D}_{I,i}) = \max_{S \in \mathcal{S}^1} \mathbb{E}_I \left\{ \mathbb{E}_{z_{1:i}} d_i(u_{\hat{I},i}; S(I)) - \mathbb{E}_{u_{I,i}} d_i(u_{I,i}; S(I)) \right\}$$

$$\geq \max_{\theta_f \in \mathcal{Q}} \mathbb{E}_I \left\{ \mathbb{E}_{z_{1:i}} d_i(u_{\hat{I},i}; f_d^i(I)) - \mathbb{E}_{u_{I,i}} d_i(u_{I,i}; f_d^i(I)) \right\},$$

Next, we would like to prove the opposite direction. Let $S^*$ be a continuous maximizer of the following objective (its existence is assumed in the proposition's statement):

$$\max_{S \in \mathcal{S}^1} \mathbb{E}_I \left\{ \mathbb{E}_{z_{1:i}} d_i(u_{\hat{I},i}; S(I)) - \mathbb{E}_{u_{I,i}} d_i(u_{I,i}; S(I)) \right\} \tag{5}$$

Since $\mathcal{I}$ is compact, by [1] there is a large enough neural network $f_d^i(\cdot; \theta_{f_d}^*)$ (with sigmoid/tanh/ReLU activation) that approximates the continuous function $S^*$ up to an approximation error $\epsilon$ (of our choice) with respect to the $L_\infty$ norm, i.e., $\|f_d^i(\cdot; \theta_{f_d}^*) - S^*\|_\infty \leq \epsilon$.

Recall that $\mathcal{D}_{\hat{I},i}$ and $\mathcal{D}_{I,i}$ are supported by compact sets. In addition, since $S^*$ is continuous over a compact set, $S^*(\mathcal{I})$ is compact as well. Let $U$ be a compact set that contains the union of the supports of both $\mathcal{D}_{\hat{I},i}$ and $\mathcal{D}_{I,i}$. Let $V$ be a compact set that contains the union of $S^*(\mathcal{I})$ and $f_d^i(\mathcal{I}; \theta_{f_d}^*)$. Since the discriminator $d_i(u; \theta_d^i)$ is a continuous function (a neural network with continuous activation functions) with respect to both $(u, \theta_d^i)$, it is uniformly continuous over $U \times V$. Therefore, we have:

$$\sup_{u \in U, I \in \mathcal{I}} \left| d(u; S(I)) - d(u; f_d^i(I; \theta_{f_d}^*)) \right| = o_\epsilon(1) \tag{6}$$

In particular, we have:

$$\max_{S \in \mathcal{S}^1} \mathbb{E}_I \left\{ \mathbb{E}_{z_{1:i}} d_i(u_{\hat{I},i}; S(I)) - \mathbb{E}_{u_{I,i}} d_i(u_{I,i}; S(I)) \right\}$$

$$= \mathbb{E}_I \left\{ \mathbb{E}_{z_{1:i}} d_i(u_{\hat{I},i}; S^*(I)) - \mathbb{E}_{u_{I,i}} d_i(u_{I,i}; S^*(I)) \right\}$$

$$\leq \mathbb{E}_I \left\{ \mathbb{E}_{z_{1:i}} d_i(u_{\hat{I},i}; f_d^i(I; \theta_{f_d}^*)) - \mathbb{E}_{u_{I,i}} d_i(u_{I,i}; f_d^i(I; \theta_{f_d}^*)) \right\} + o_\epsilon(1)$$

$$\leq \max_{\theta_{f_d} \in \mathcal{Q}} \mathbb{E}_I \left\{ \mathbb{E}_{z_{1:i}} d_i(u_{\hat{I},i}; f_d^i(I)) - \mathbb{E}_{u_{I,i}} d_i(u; f_d^i(I)) \right\} + o_\epsilon(1)$$

which completes the proof. $\square$

## 2 Training

### 2.1 Architecture

We use ResNet-34 for the hypernetworks of both the generator and discriminator, with an embedding size of size 512. A multi-head dense linear layer is then applied and projects the image embedding into the different convolutional blocks of the main network. The main networks, (i.e., the generator and discriminator) share the same architecture and consist of 5 conv-blocks per scale of the form Conv(3 x 3)-LeakyRelu with 64 kernels per block. For the generator, we hold a set of 10 linear heads projections for each scale, where each projection outputs the weights (or the biases) of its respective

scale in the generator. For the discriminator, when training, only the current scale's linear projections are needed, thus we hold a single set of 10 linear head projections.

All LeakyReLU activations have a slope of 0.02 for negative values except when we use a classic discriminator for single image training, for which we use a slope of 0.2. Additionally, the generator's last conv-block activation at each scale is Tanh instead of ReLU and the discriminator's last convolutional block at each scale does not include any activation.

Differently from [3], but similarly to [2] we do not gradually increase the number of kernels during training.

Grouped convolutions were used in order to perform parallel computations in the main network for each image with its respective weights to speed up training.

## 2.2   Progressive training

We train with an initial noise of width $s_0 = 25$ pixels, except when training on the 50-images dataset and for the single mini-batch experiment (for which we use $s_0 = 28$ and $s_0 = 27$ respectively) such that the dimensions of the initial noise are $(\lceil s_0 * ar \rceil, s_0)$ where ar is the aspect ratio of the image. If trained with multiple images, the default aspect ratio used for training is 3/4.

In terms of sizes of the images processed at each scale, we progress in a geometrical way as [3] with a scale factor of $r = 0.6$ i.e., at each scale $i > 0$ ,images of size $s_i = \frac{s_{i-1}}{r}$ are processed. This results in 7 scales for an image of size 256. Although we train in a progressive manner, our hypernetworks receives as input (128,128) constant sized versions of the real images regardless of the current scale processed. We progress from a scale to another at the end of the training of a current scale in the generator by copying the weights of its 10 linear projections to the next scale's projections and freeze the current scale, for the discriminator, we simply copy the weights of its linear projections and can safely delete its current set of linear projections from memory.

## 2.3   Optimization

The loss function is minimized using Adam optimizer with momentum parameters $\beta_1 = 0.5$, $\beta_2 = 0.999$ and different learning rates for each training setting, which we decreased by a factor of 0.1 after 80% of the iterations. We used $\lambda_1 = 0.1$ and $\lambda_2 = 50$ for the coefficient of the gradient penalty in WGAN and the reconstruction loss respectively, $\lambda_2 = 10$ can also be used and yield good results. We clip the gradient s.t it has a maximal L2 norm of 1 for both the generators and discriminator. Batch sizes of 16 were used for all experiments involving a dataset of images.

|  | $\mathrm{lr}_g$ | $\mathrm{lr}_d$ |
|---|---|---|
| Single image | 1e-5 | 5e-4 |
| Single mini-batch | 1e-5 | 1e-5 |
| Dataset | 5e-5 | 5e-5 |

Similarly to [3], we use MSE as the reconstruction loss, and at each iteration we multiply each noise map $z_i$ for $(i > 1)$, by the RMSE obtained. This results in zero-mean and MSE varianced gaussian distributed noise maps and indicates the amount of details that need to be added at that scale for the current batch. For the reconstruction, a single $z_1^0$ fixed random noise is used for all the images.

For feedforward modeling and applications, we use this single fixed random noise, and we multiply each scale's intermediate noise map by the RMSE obtained at the last iteration of this same scale during training.

### 2.4 Number of iterations

|  | number of iterations by scale |
| --- | --- |
| Single image | 1500-2000 |
| Single mini-batch | 2000 |
| Places-50 | 4000 |
| LSUN-50 | 5000 |
| C250 | 20000 |
| C500 | 30000 |
| V500 | 25000 |
| V2500 | 100000 |
| V5000 | 150000 |

### 2.5 GPU usage for training models

|  | GPU memory usage (256x256 resolution) |
| --- | --- |
| Single image | 11GB |
| Single mini-batch | 11GB-15GB |
| Datasets | 22GB |

At test time, the GPU memory usage is significantly reduced and requires 5GB. We trained all of our single image models and baselines with a single 12GB GeForce RTX 2080. The other models were trained on a single 32GB Tesla V100. Notice we compared the single and the dataset runtimes in Table 1 in the main paper, by approximating the runtime on a GeForce, training until scale 5 on GeForce RTX 2080 (until 12GB is out of memory) and by taking in account the difference in power between the latter and V100 GPU.

## 3 Training with a pretrained image encoder

In this section, we consider training our method with a "frozen" pretrained ResNet34 i.e., optimizing only the linear projections.

Our method uses single linear layer projections, which strongly restricts the expressiveness of our network if the image encoder is "frozen". We thus experimented with increasing the depth of these projection networks.

Below are the results on Places-50 :

|  | End-to-end (our setting) | 1 layer | 3 layer | 5 layer |
| --- | --- | --- | --- | --- |
| SIFID | 0.05 | 0.26 | 0.14 | 0.17 |
| mSIFID | 0.07 | 0.56 | 0.23 | 0.27 |
| Diversity | 0.50 | 0.79 | 0.63 | 0.63 |

If the problem could be learned with a "small enough" depth, our method would benefit from even faster training, at the cost of enlarging the model (and its consequences on inference time). Even though the results are convincing (both visually and quantitatively) in favor of end-to-end training, we prefer not to reject the hypothesis that proper hyper-parameter tuning and perhaps some adaptations could lead to decent results with a frozen backbone.

## 4 Single-Image Generation

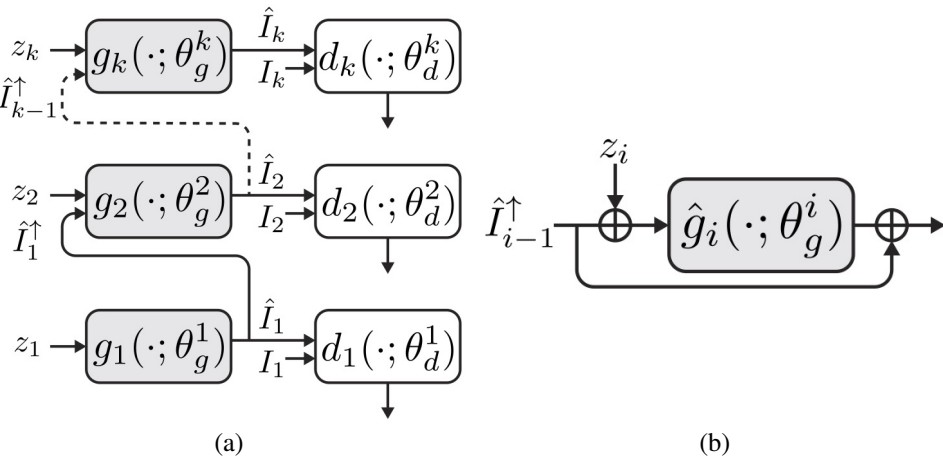

Figure 2: **Single-Image model architecture. (a)** The complete hierarchical structure of generators and discriminators. **(b)** The inner architecture of $g_i$, consists of noise addition and residual connection.

Fig. 2 illustrates the single-image architecture with the internal skip connection, of [3], as we discuss in section 2.

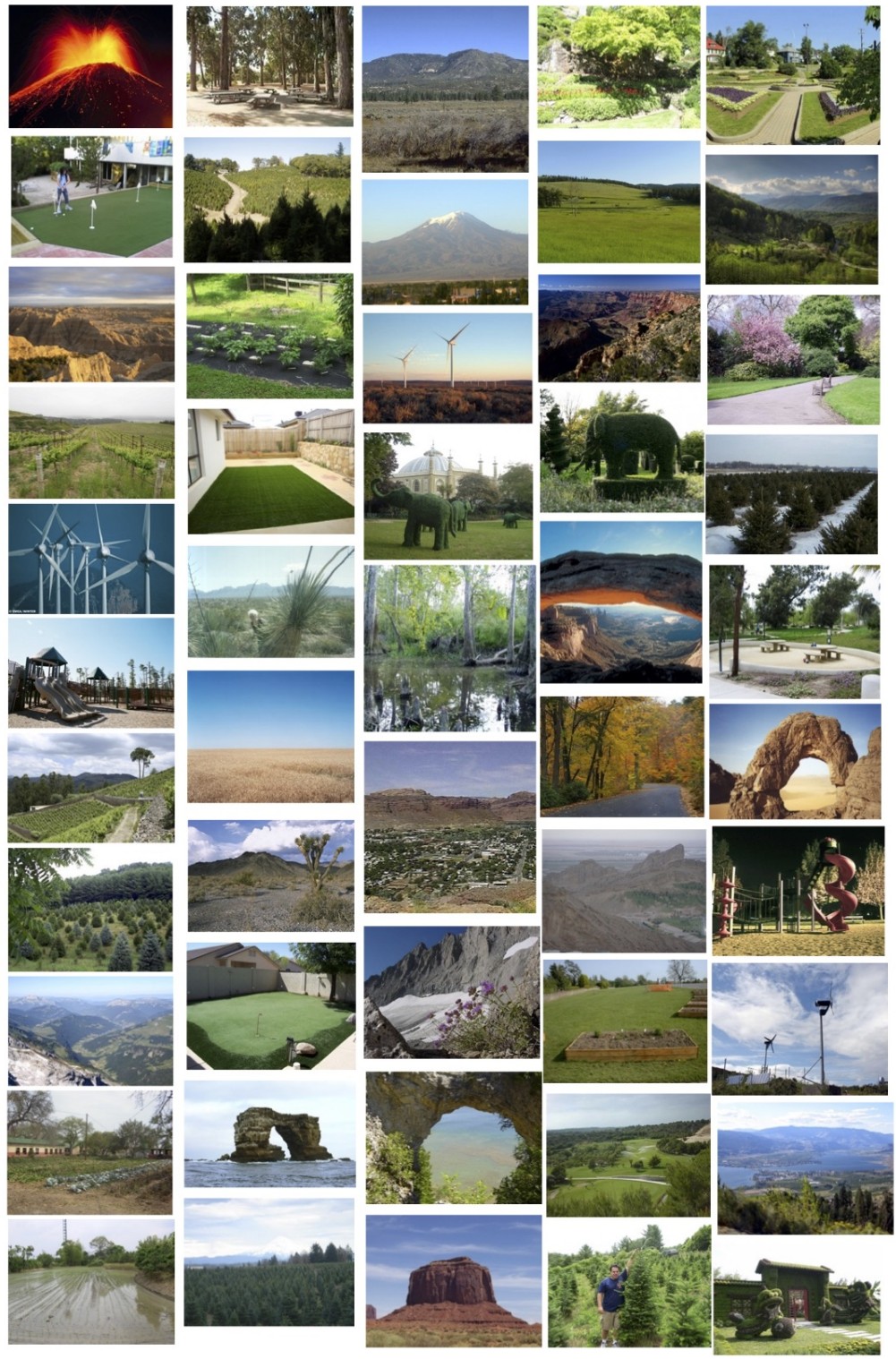

Figure 3: **Places-50 real images** : The dataset used in SinGAN

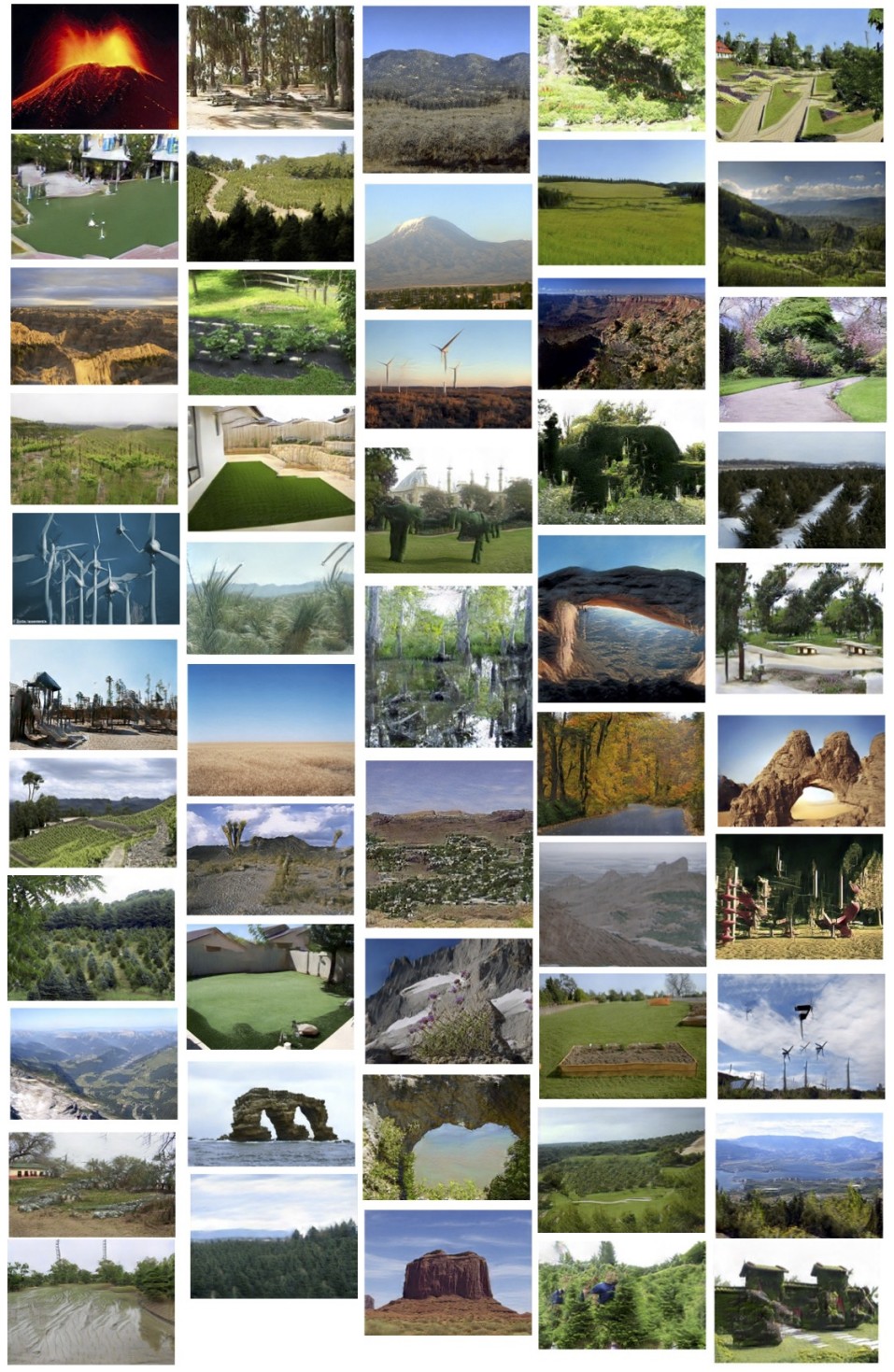

Figure 4: **Places-50 image generation:** Results of our model when training on a single image one by one, similarly to previous methods.

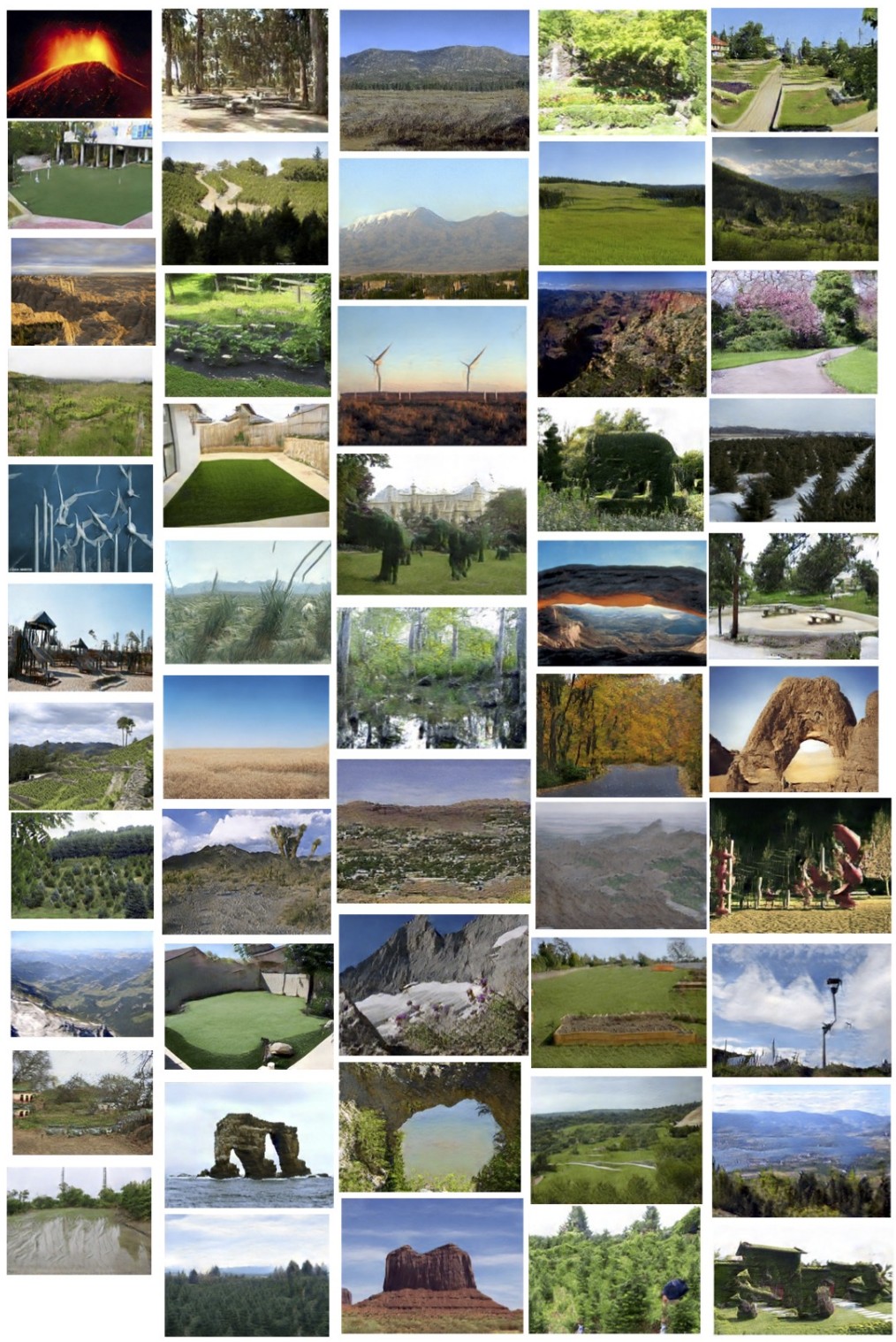

Figure 5: **Places-50 image generation:** Results of our model when training on the 50 images altogether as a dataset.

Fake images – Single training

Original image

Fake images – Dataset training

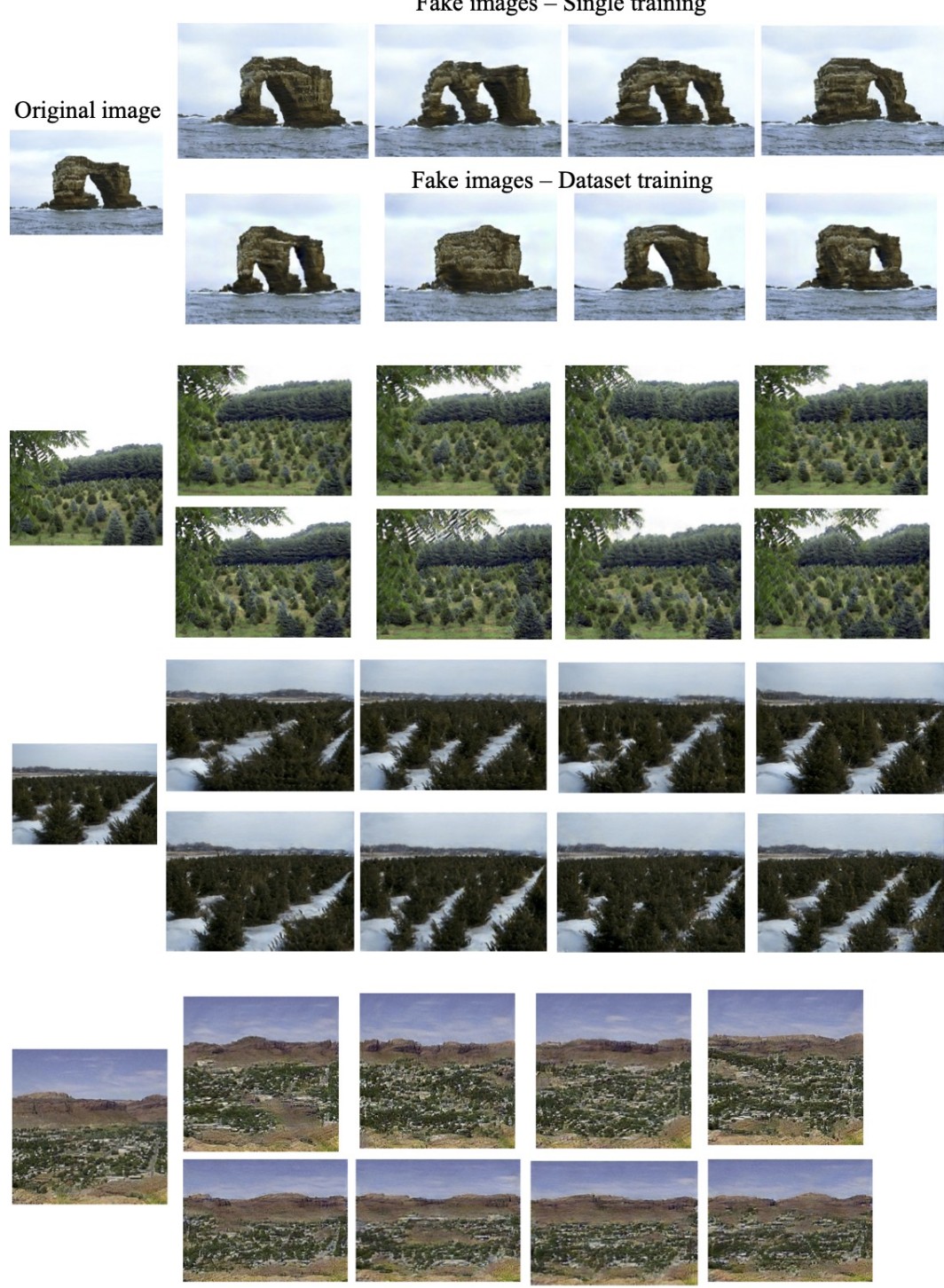

Figure 6: **Places-50 image generation:** Comparison between single image training and dataset training.

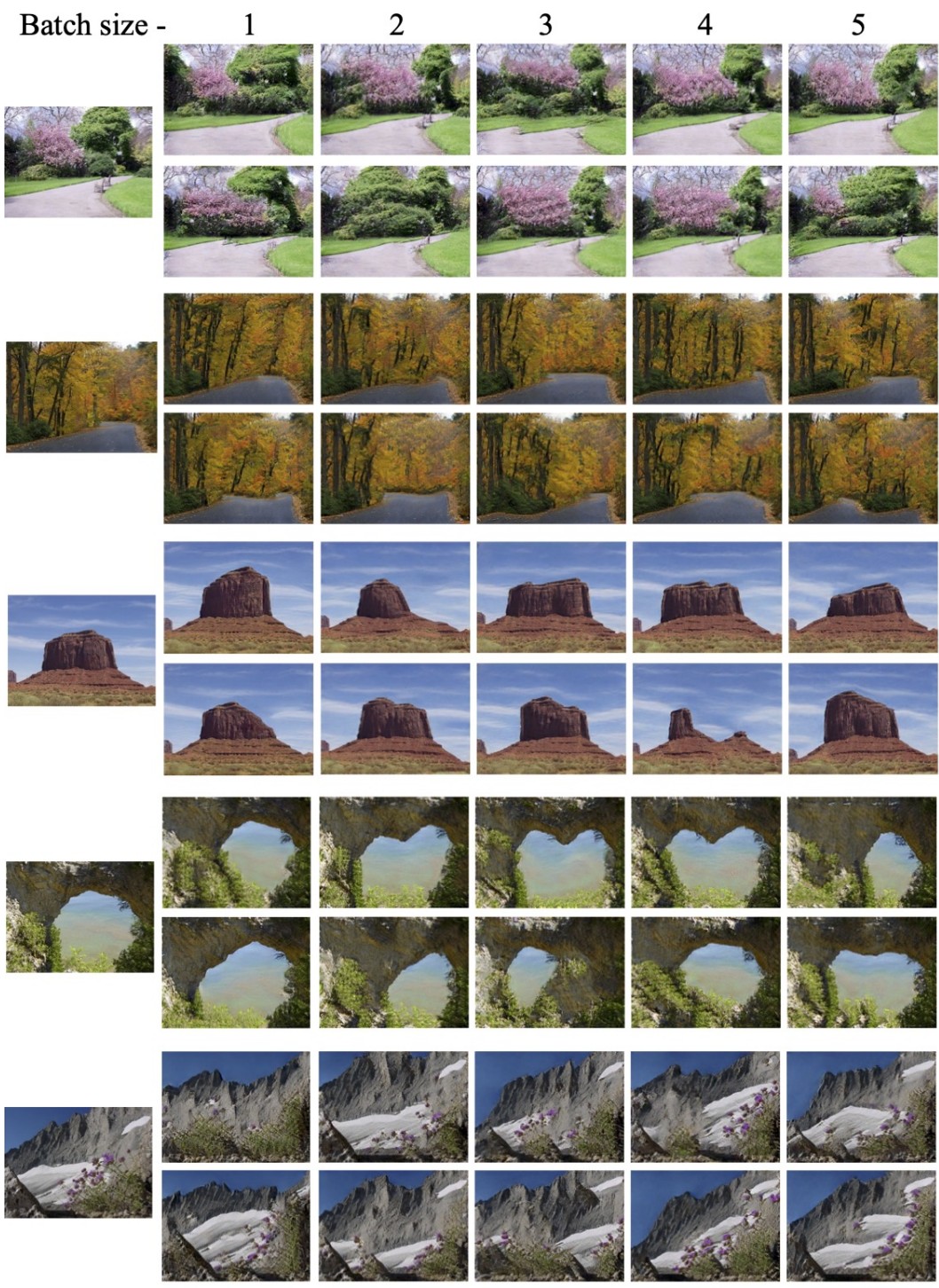

Figure 7: **Places-50 image generation:** Comparison between different batch sizes when training in a single mini-batch scheme. As can be seen, our method yields realistic results with any batch size.

Original image                                    Random samples

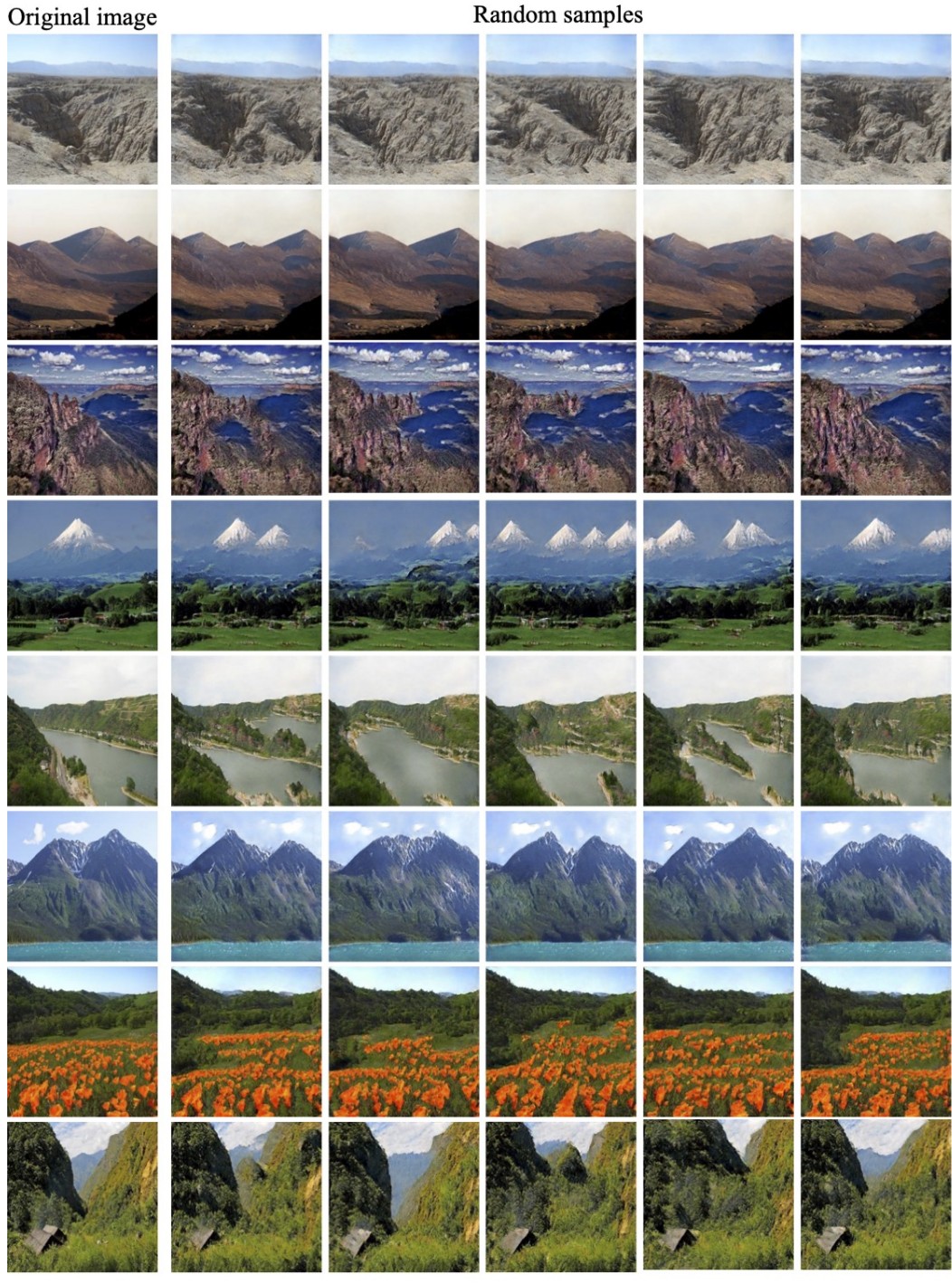

Figure 8: **Valley-500 image generation**

Original image          Random samples

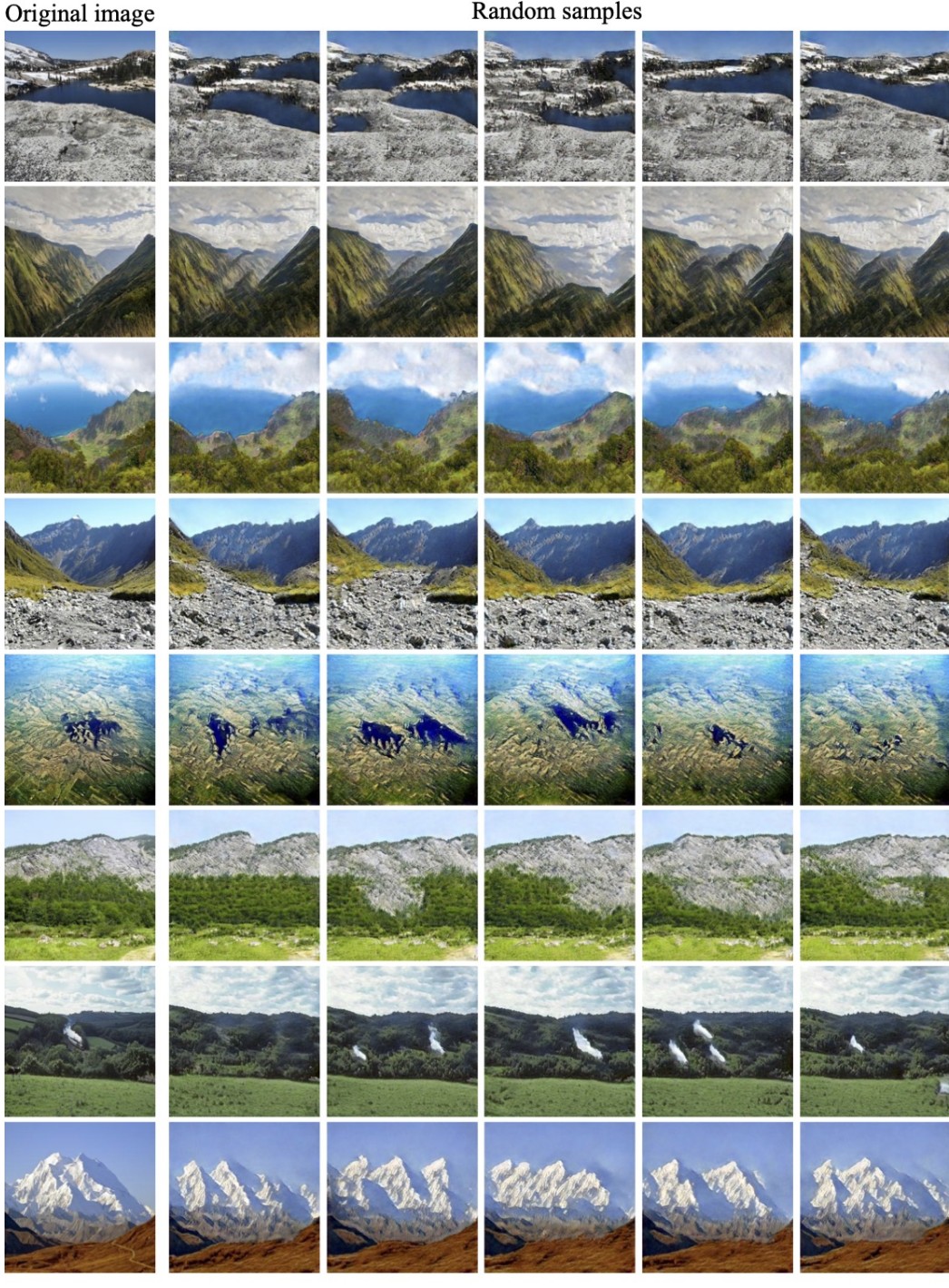

Figure 9: **Valley-2500 image generation**

Original image                                    Random samples

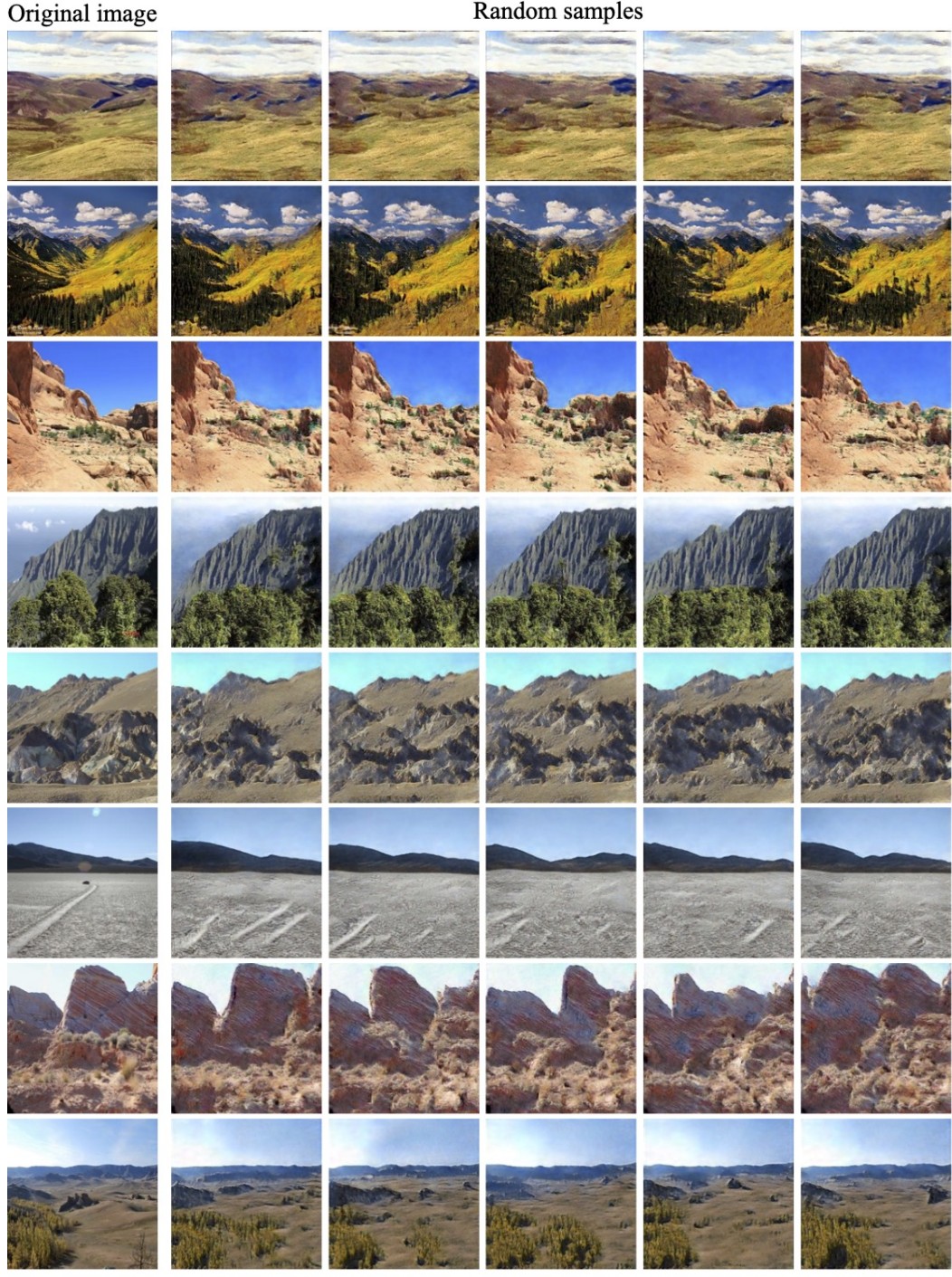

Figure 10: **Valley-5000 image generation**

# 50 images – LSUN dataset

Real                       Ours                          SinGAN   ConSinGAN

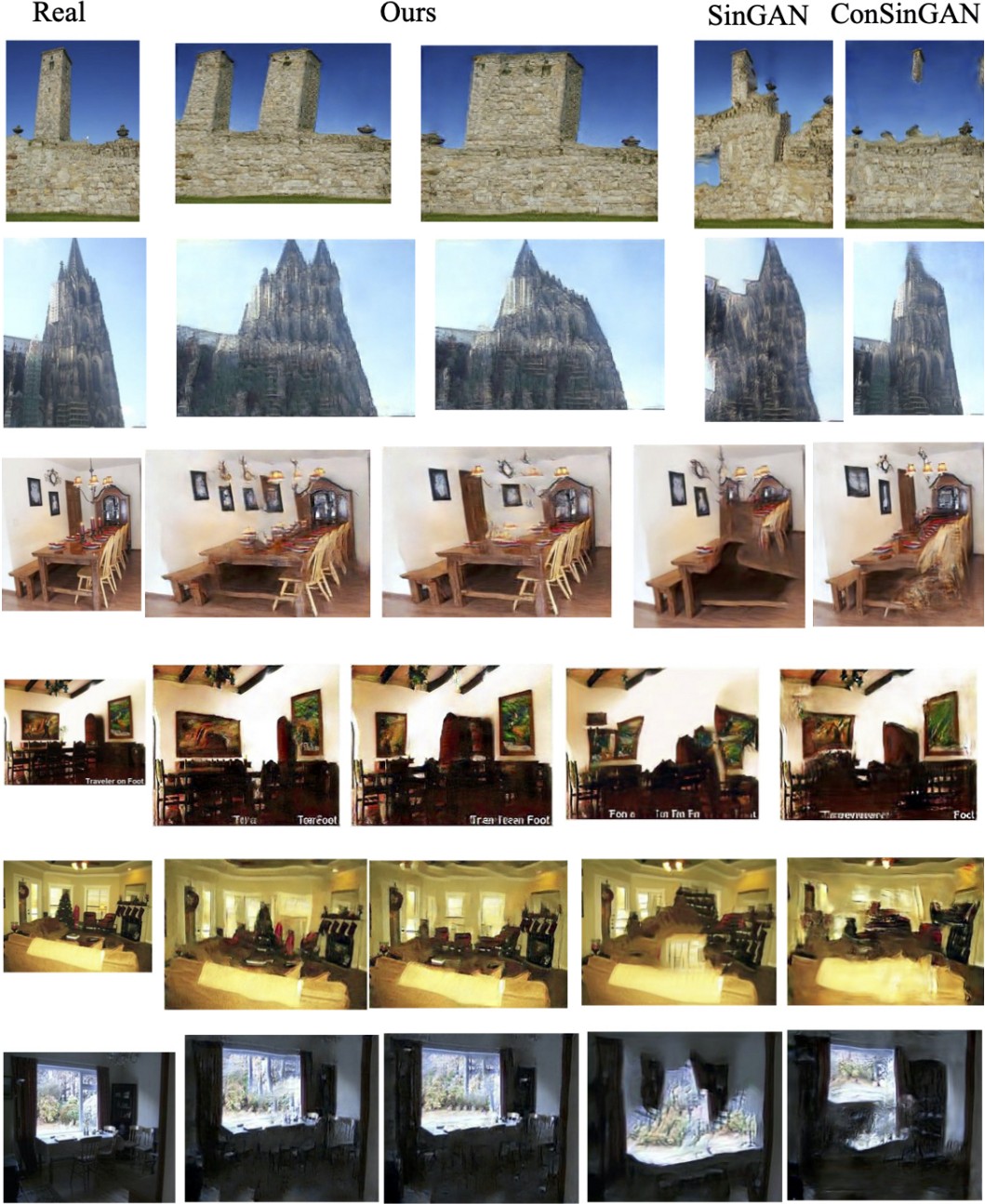

Figure 11: **LSUN-50 image generation:** Comparison with baselines. Our approach yields significantly more realistic results when trained on images with complex structures.

250 images sampled from Churches Outdoor category - LSUN

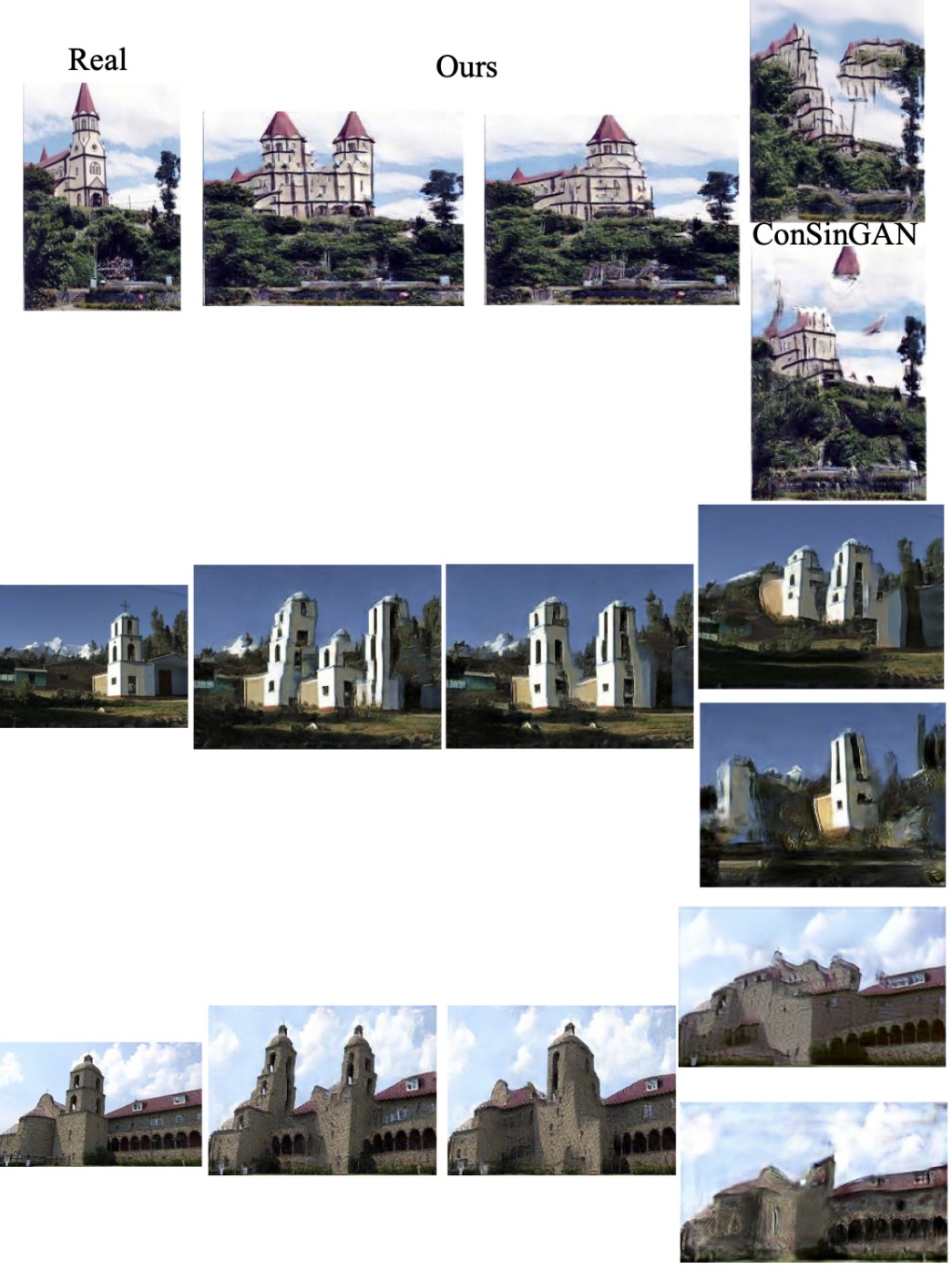

Figure 12: **Churches-250 image generation:** Comparison with baselines. Our approach often succeed to create realistic and crisp images even when single image alternatives fail.

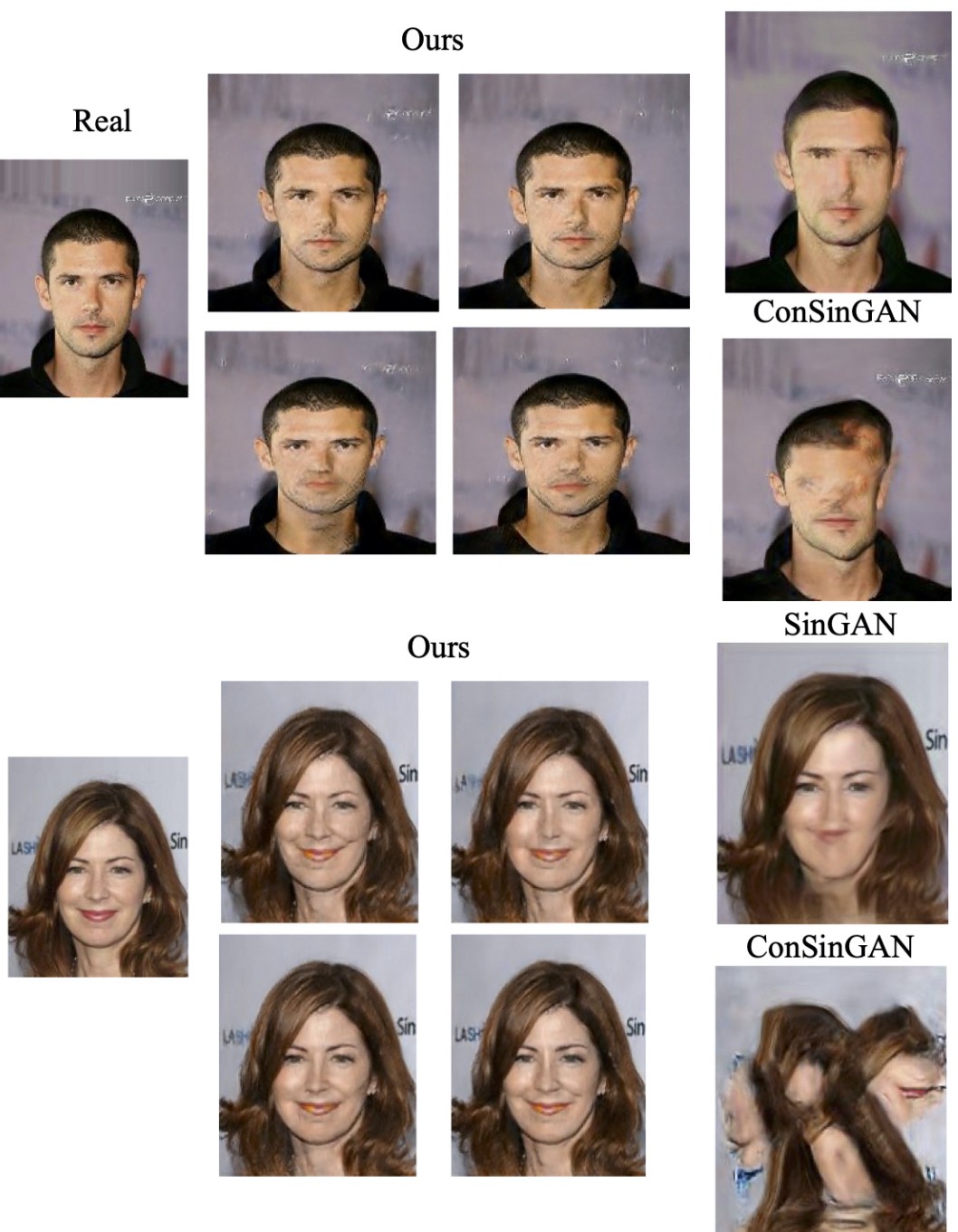

Figure 13: **50 randomly sampled images generation - CelebA dataset:** Comparison with single image alternatives, where we used the same initial noise size (of width 22) for all of the methods to allow for a fair comparison. Even though our method generates more realistic images than baselines by a notable margin, our results are still non-comparable to classic face generation (by standard GAN and Flow-based models). Thus, we consider face datasets as a limitation of our approach.

## 5  Editing, Harmonization and Animation

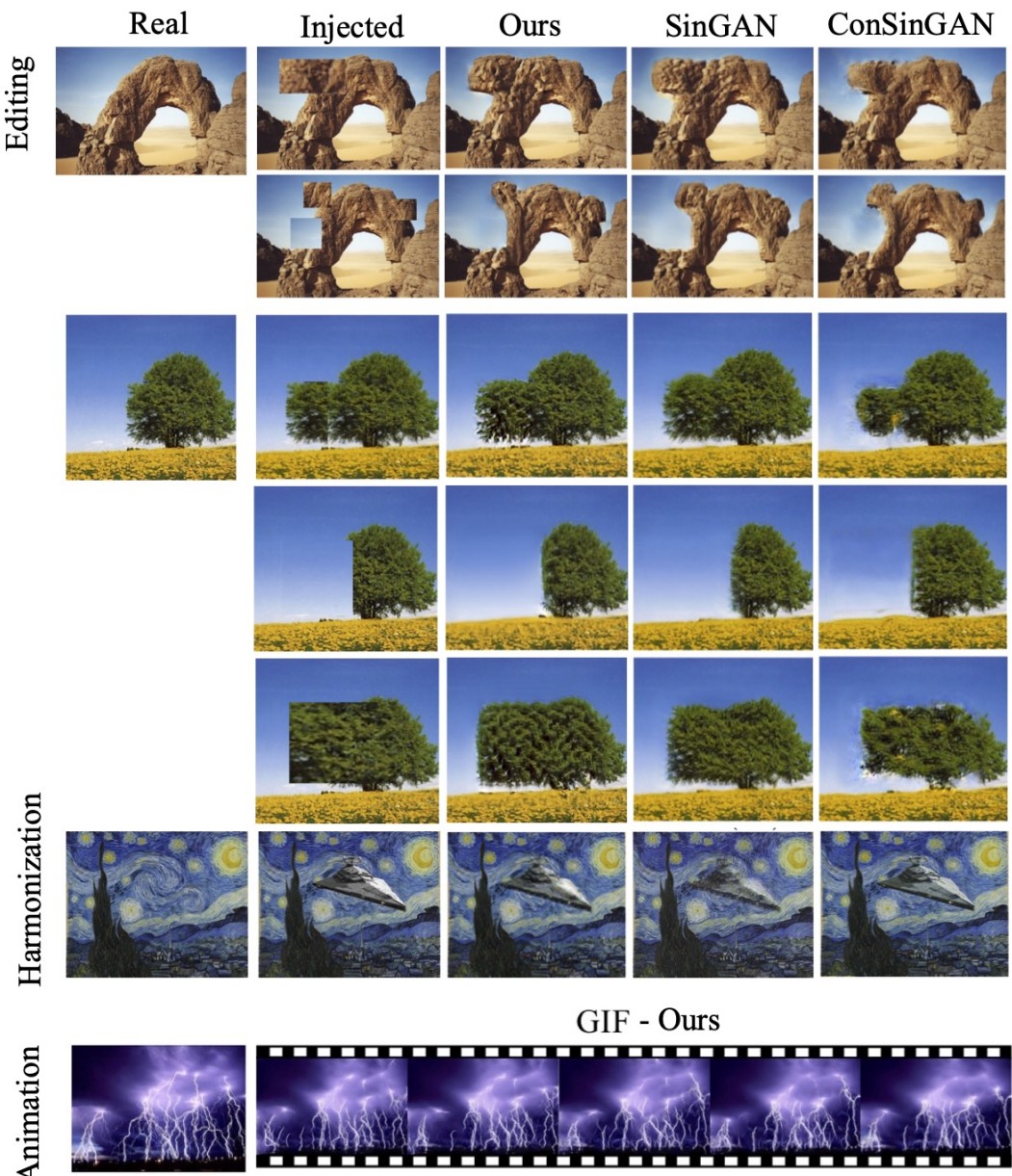

Figure 15: **Editing, Harmonization and Animation:** Comparison with baselines. Our method allows to efficiently train one unified model for all images and various tasks. Results were obtained when training on these images along with Places-50 dataset.

Original image

Single mini-batch training

Arbitrary sized and aspect ratio generated images

V500 dataset training

Figure 14: **Randomly sized image generation:** Due to the fully convolutional architecture adopted, all of our models are able to generate an image with an arbitrary size of aspect ratio by simply changing the dimensions of the noise maps used. The results shown were obtained with single mini-batch training and V500 dataset training.

# 6 Interpolation

We conducted an experiment to study the smoothness of our interpolations at different scales. We estimated the slope of the generated images $H_i(\alpha)$, for a fixed set of random seeds, on a discrete set of values $\alpha \in \{0.1j\}_{j=1}^{9}$ as follows: $s_{i,j} := \frac{\|H_i(\alpha_{j+1}) - H_i(\alpha_j)\|_1}{h \times w \cdot (\alpha_{j+1} - \alpha_j)}$, where $h \times w$ is the size of the images. As can be seen in Fig. 16, the interpolations at higher scales tend to be significantly smoother than the interpolations at lower scales.

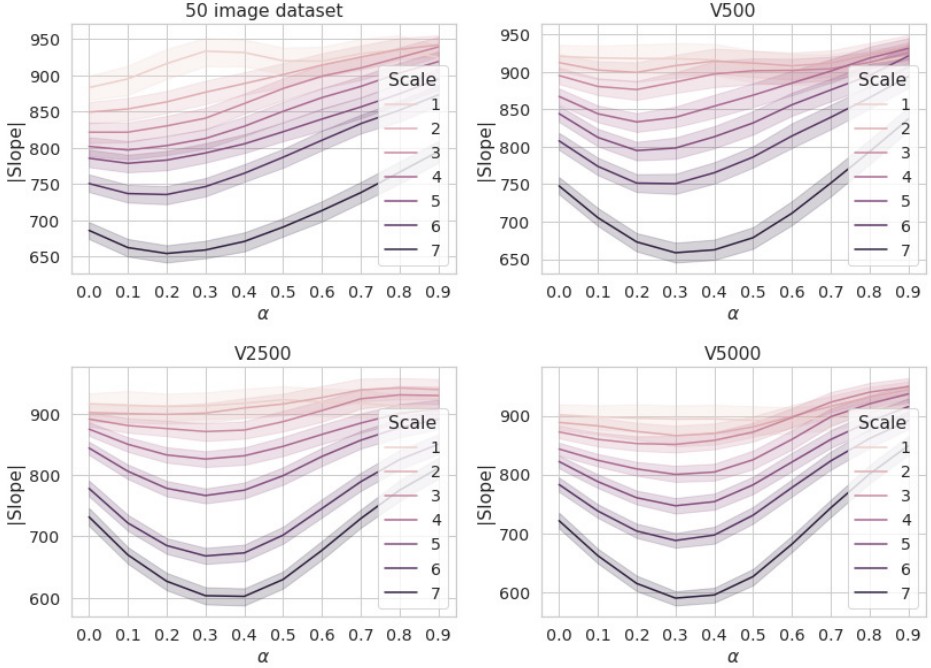

Figure 16: **Smoothness rate of the interpolations.** We plot the smoothness rate $s_{i,j}$ (y-axis) as a function of $\alpha$ (x-axis), averaged over 500 pairs of images $A, B$ along with their standard deviations.

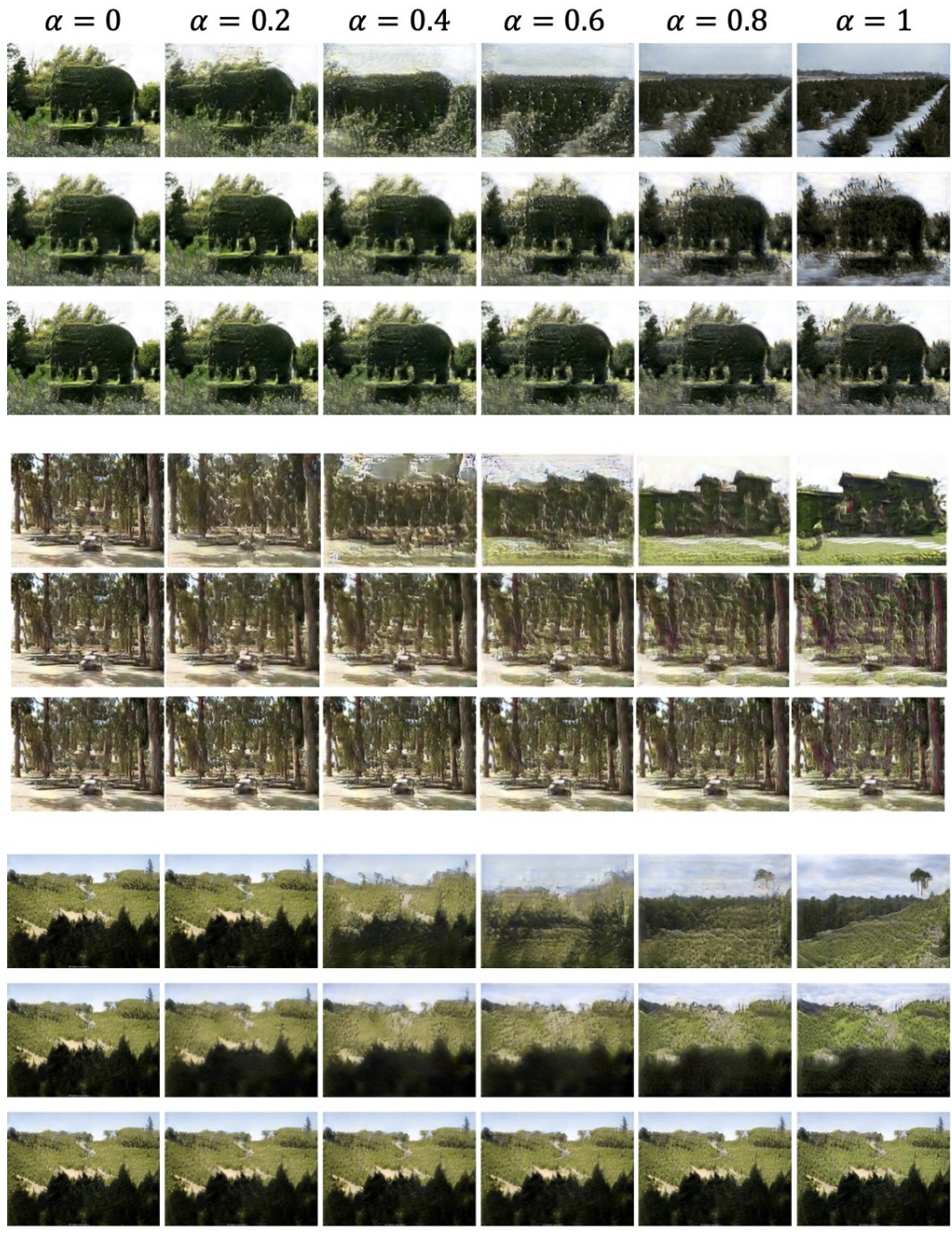

Figure 17: **Interpolation - Places-50:** Results show different generated images when injecting through the first, middle and last scale of the model.

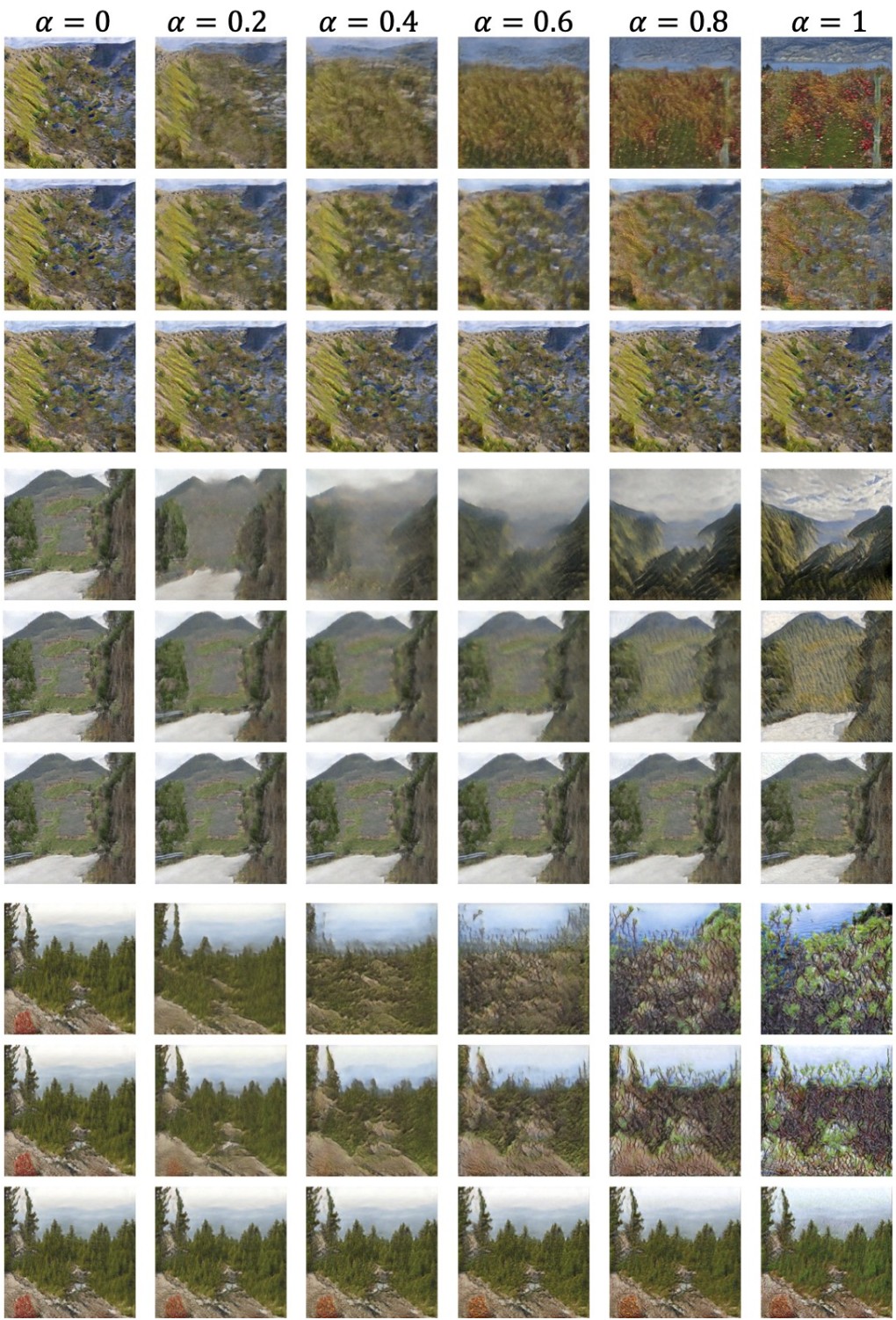

Figure 18: **Interpolation - Valley-500:** Results show different generated images when injecting through the first, middle and last scale of the model.

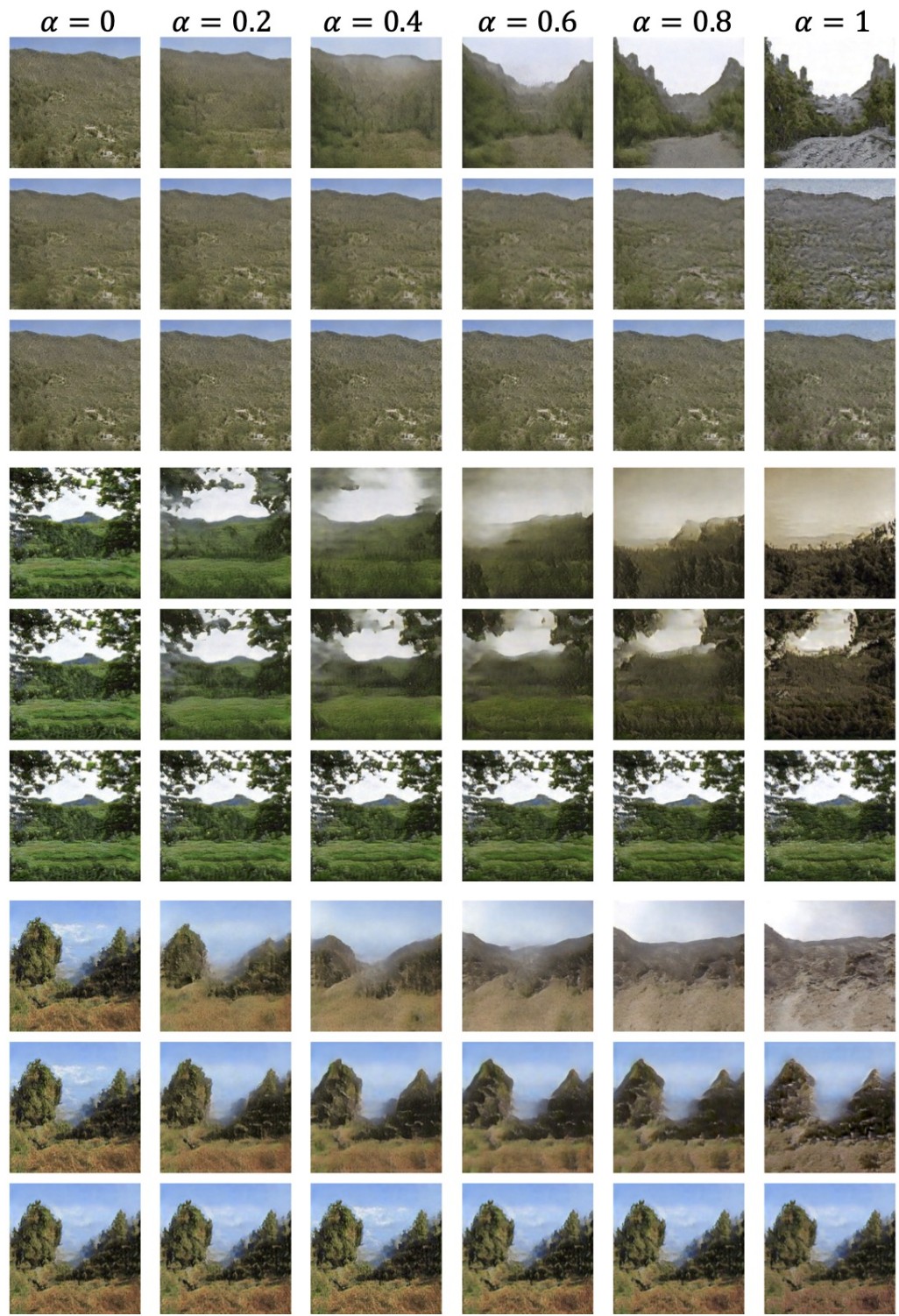

Figure 19: **Interpolation - Valley-2500:** Results show different generated images when injecting through the first, middle and last scale of the model.

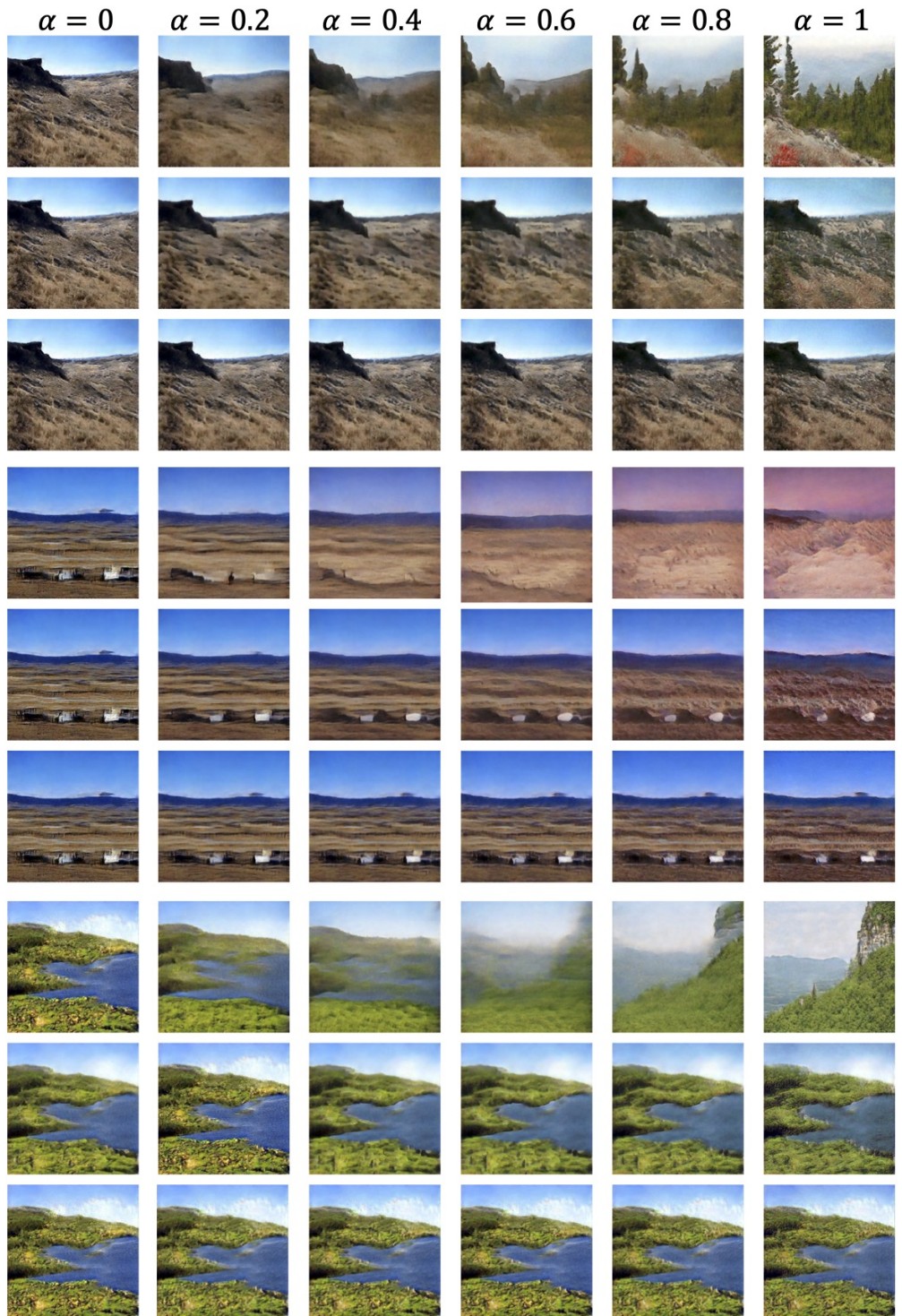

Figure 20: **Interpolation - Valley-5000:** Results show different generated images when injecting through the first, middle and last scale of the model.

# 7 Feedforward modeling

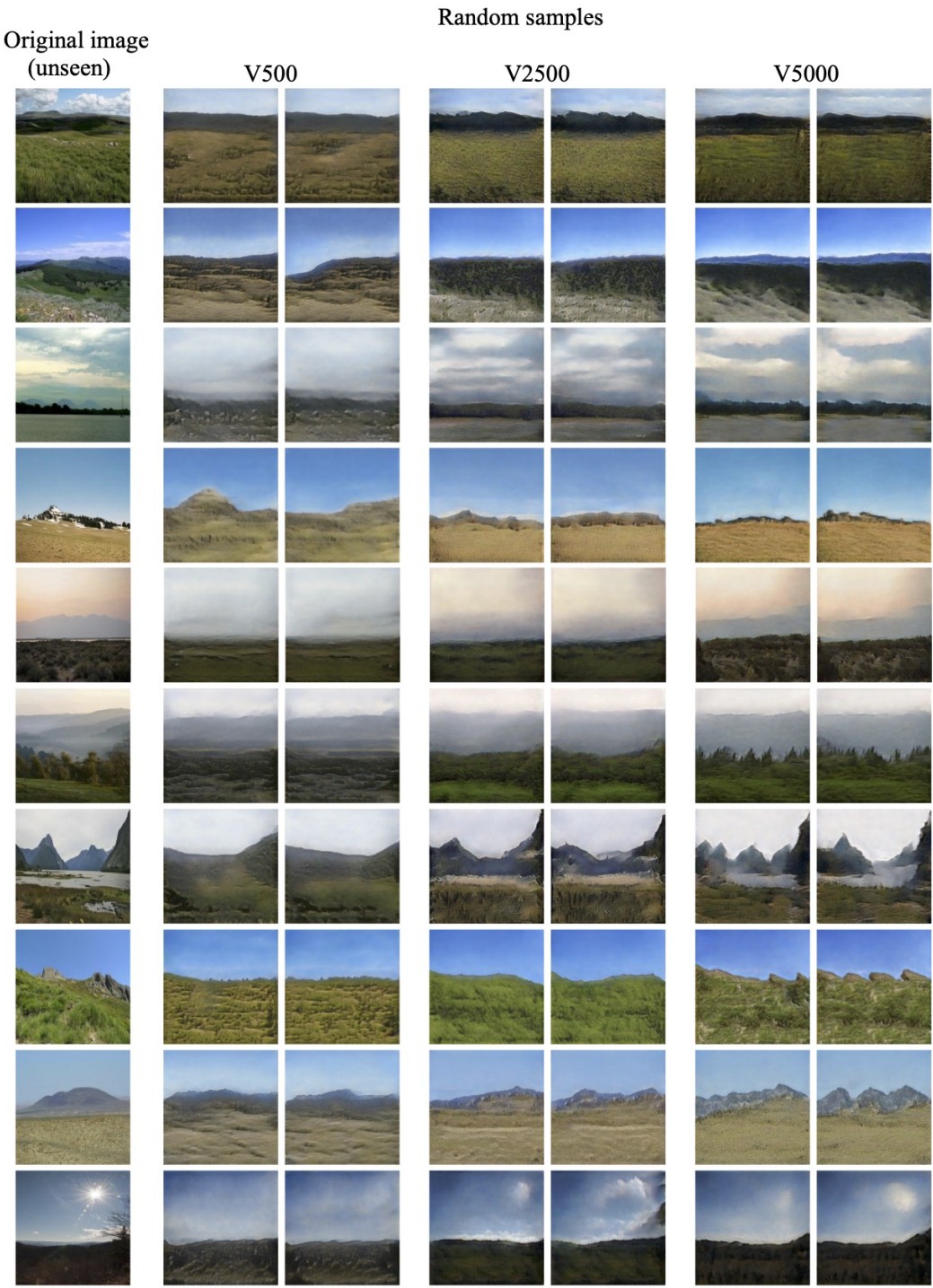

Figure 21: **Feedforward modeling:** Our meta learning approach allows us to feedforward an unseen image through our model in a fraction of a second and generate coherent results. Results were obtained using the 100 validation set from Valley category of Places dataset, when trained on the train set of Valley-5000 (the whole category).