# OpenReview forum: "Meta Internal Learning"
_NeurIPS.cc/2021/Conference — NeurIPS 2021 Poster_

### Official Review · Reviewer_FnNF · 2021-07-15

**Rating:** 6
**Confidence:** 4

**Summary:**

This paper introduces an advanced version of SinGAN by incorporating the hypernetwork into the framework so that the proposed model can perform all the tasks that the original SinGAN can do, but with much faster test inference.

**Ethical Concerns:**

No comment

**Limitations And Societal Impact:**

No comment

**Main Review:**

The idea of employing hypernetwork to train both generators and discriminators is very interesting, and the motivation of fast inference is convincing enough for me.


My main concern is on the quality of the results (Figure 3 and 4). I partly agree (but with some conditions) that if there is no shortage of unlabeled images, single-image GAN is unjustifiable. This must go with conditions that the test image performance is better than or at least comparable to the original SinGAN.


Zooming at Figure 4, however, the quality of the results does not seem to match with SinGAN (even on train data with the model trained on the larger dataset!). For example, the V5000 results does not look crisp nor realistic when looking at the boundaries of the buildings. These qualitative results do not match with the quantitative results shown in Table 1 and Table 3. This needs more explanations.


In Table 3, using a smaller dataset gives better results on training data, and this trend is opposite in the test data. What would be the explanation for the former case? (+ why doesn’t this go along with the qualitative results in Figure 4?)


Speaking of the results, I don’t see the benefit of the availability to interpolate between single image GANs (Figure 3). I understand that the framework allows this to perform and indeed this is new, but cannot see what kinds of advantage it would bring by doing so. This needs more discussion, or reduce the weight it takes in the paper (Currently, it is mentioned with the other major contributions).


In addition, one of the contributions that the authors argue is that they are the first (as far as they know) to perform adversarial training with hypernetworks. In fact, there exists a very similar work that did use hypernetwork for generator and train it using the adversarial loss: “Generative Models as Distributions of Functions”. Although it does not use the hypernetwork scheme for both generator and discriminator, this is noteworthy. However, I do not think this compromises the novelty much since the work is quite recent and has not been published anywhere yet.


The organization of the paper can be also improved. For example, in my personal opinion, it looks better to spare more space in Backgound section for introducing the intuition behind the SinGAN and its characteristics than providing the technical details (like eq. (2), (3), and (4)), which are not important to understand the main idea and context.


Overall, the idea is interesting and the advantages of using the framework are clearly seen. However, since the results do not go along with the first two benefits (Line 24 – Line 27), which are very critical, it is hard to give a strong score to the paper.

---

Update: Raised score to 6, please see the comment.

---


**Time Spent Reviewing:**

10

---

> ### Author Response · Authors · 2021-08-10
> **Author response**
>
> Thank you for your comprehensive review.
>
> Figs. 3 and 4 provide single examples, solely provided for illustration purposes. We kindly ask the reviewer to avoid drawing conclusions based on these. We refer the reviewer to the supplementary material and to the following links: [Link 1](https://imgur.com/a/TYNQfSc) , [Link 2](https://imgur.com/a/0xpMcVu) , [Link 3](https://imgur.com/a/fhSCQXE), which contain many real-sized samples on datasets requested by the other reviewers. Our method generates sharp, realistic, and diverse results (for any dataset size) and outperforms baselines both quantitatively and qualitatively.
>
> Specifically, regarding Fig. 3, we focus on the effect of interpolating at different levels (lines 275-280). For illustrative purposes, we selected two very different images such that the differences would be clear.
>
> Regarding Fig. 4, the text in L281-275 focuses on the need for a larger training dataset in order to obtain better results. We selected an image where the differences, which are clear also in Tab. 3, are visible. When comparing with single image methods, the text focuses on training time for a new image as part of a dataset.
>
> For comparisons on single image training, we refer to Appendix C, where many samples are presented. As can be seen, the quality of our outputs is very good, as well as the performance on different applications.
>
> Table 3 training vs testing -- the behavior, e.g., of diversity given a training image is different from that of a test image, as can be seen in the table. On the train set, all methods perform relatively the same, except that mSIFID is better for the smaller training set, this can be explained by the fact that we did not control for the training time per image, which is much larger for the smaller datasets, leading to additional overfitting and better results on the training set.
>
> On the test set, the importance of a larger, more diverse training set is apparent across the scores. This is not surprising, since the larger the training set, the more coverage of the visual space is given. Therefore, given a new image, modeling is expected to be better for larger datasets, leading to an improvement in the scores.
>
> When working with images, interpolation has a few entertainment/creativity applications, can be used by designers, and can teach us about the underlying working on the trained networks. One can also use the creation of virtual samples for improved training using methods such as mixup.
>
> Other work on hypernetwork GANs -- the work noted by the reviewer differs from our work in two main directions, first, it does not apply meta learning, meaning the latent space learned by the model is dependent on the learned dataset, and second, it does not apply internal-learning.
> As stated by the reviewer, unpublished work put on arxiv less than 3 months before the submission deadline cannot affect the novelty much. However, it does point to the timeliness of our work. As also mentioned by the reviewer, we are unique in applying hypernetworks to both the generator and the discriminator. This is a non-trivial feat that requires a theoretical analysis that is entirely novel.
>
> The theoretical analysis required some technical background, which made the background section a bit drier than usual.
>
> In lines 24--27 we mention four potential benefits: (i) knowledge and feature sharing between the different images, (ii) better define the boundaries between the distribution obtained from the input image and those of other images, (iii) possibility to avoid the costly training phase given a novel image, and employ feedforward inference instead, and (iv) mixing different single-image models to create novel types of images.
>
> Point (ii) means that modeling of structure, for example, improves with more images. This is apparent in the test metrics of Tab. 3 and also in the new LSUN experiments requested by Reviewer 1LvZ. Point (iii) is evident in our feedforward inference results. Point (iv) is acknowledged by the reviewer.
>
> Point (i) is indeed a technical point that describes how networks are training with multiple samples and it contrasts single image training. It will be rephrased to make this clearer.

---

> > ### Comment · Reviewer_FnNF · 2021-08-19
> > **Thanks for addressing my points.**
> >
> > After looking at the authors' rebuttal, I raise my score to 6. However, as the other reviewers have mentioned, I strongly suggest the authors consider re-organizing the manuscripts (at least partly); e.g., to put off some of section 2 to the appendix and bring more results to the main manuscript. I would also like to ask, if possible, to find better images for illustration. By just looking at the current images, it looks very off with the conclusion stated in the main content.

---

> > > ### Author Response · Authors · 2021-08-19
> > > **Thank you for your message**
> > >
> > > (our posts crossed...)
> > >
> > > We appreciate your reevaluation and would reorganize Sec 2 according to your suggestion. We would also replace the images used for illustration.

---

### Official Review · Reviewer_1LvZ · 2021-07-16

**Rating:** 6
**Confidence:** 5

**Summary:**

The paper proposes a method to learn hypernetworks, trained on a larger dataset (500-5000 images), for single-image GANs. These hypernetworks are then used to initialize the GAN for a specific single image to speed-up training time and improve image quality/diversity. The single-image GAN follows the same architecture as SinGAN and the individual generators and discriminators for a specific image are initialized by the trained hypernetwork and then fine-tuned for the specific image. The resulting single-image GANs outperform related work (SinGAN, ConSinGAN, ...) based on SIFID, diversity, and training time.

**Limitations And Societal Impact:**

The authors address limitations and societal impacts.

**Main Review:**

The idea of using a larger dataset to speed up the training of single-image GANs on a single image is promising. Using hypernetworks and meta-learning may be one suitable approach for this. The method in this paper seems to deliver promising results while only needing 5 minutes of fine-tuning for a given image.

It would be more interesting to see how well this approach performs on more complex images and how much it benefits from large-scale training in this case. SinGAN-like approaches are not good at modeling complex structures/objects and it seems to me that using hypernetworks would be most helpful in these cases. Landscapes do not contain many complex structures so it would be better to test this approach on other datasets, e.g. LSUN churches or something similar.

Also, is it necessary to train an image encoder as you do given that there are many pre-trained image encoders already available? Have you tried using a pretrained (frozen) image encoder as conditioning for your hypernetworks?

While I get that your approach supports applications that are not supported by traditional single-image GANs (e.g. image interpolation and mixing) I think it would still be beneficial to compare against at least some of the original applications directly in the main paper (unconditional synthesis, harmonization, editing, retargeting, ...). At the moment this is all hidden in the supplementary which makes it difficult to directly qualitatively compare your approach to the baselines.

I believe some (most) of section 2 (2.1 can be summarized in one paragraph like 2.2) can be moved to the supplemental to free up more space to run more experiments and evaluations. Similarlay, section 3 can be shortened (e.g. equations 9-11 are a repetition of equations 6-8).

Overall, I believe this paper would benefit from a more thorough analysis of the approach on different datasets (that also contain more complex structures than landscapes), more direct comparisons to the baselines on some applications (generation, harmonization, editing, ...), and possibly running a user study to get direct comparisons between the different models besides SIFID/diversity (both of which have been shown to have weaknesses).

***
Update: Raised score to 6, see comment.
***

**Time Spent Reviewing:**

2.5

---

> ### Author Response · Authors · 2021-08-10
> **Author response**
>
> Thank you for the comprehensive review.
>
> We accept your suggestions regarding shortening a few sections and exchanging results with the supplementary. Since our paper contains a significant theoretical analysis with important practical implications, which also necessitates some background text, some sacrifices have to be made with respect to presenting results in the main text.
>
> Following the review, we have applied the method to two subsets of the LSUN dataset. The reviewer’s hypothesis, which we gladly adopt,  is that datasets with more structure would further support our method since the structure is ambiguous from a single image.
> First, in order to provide a fair and fast comparison with baselines, we trained on the 50 LSUN images used in ConSinGAN, which consists of a small sample of images from each category. As can be seen in the following [Link](https://imgur.com/a/0xpMcVu), even when using 50 images, our method outperforms baselines by a significant margin on these complex structures.
>
> We also trained on the 250 first images of the church outdoor category - see [Link](https://imgur.com/a/TYNQfSc) . Again, our method generates significantly more realistic samples while reducing the runtime (by image) by a factor of 10. (5 minutes vs 50 minutes)
> Regarding side by side comparison of unconditional synthesis, the samples in Sec. C2 and C3 are provided similarly to SinGAN to allow for a direct comparison. We will add these results and provide side-by-side comparisons for applications, as requested.
>
> Using a pretrained encoder has naturally been considered. Since our work is the first (up to the submission date) to use a full hypernet adversarial framework, we prefered to stay as close as possible to our domain of application - internal learning, which as far as we know, do not use any supervision or prior knowledge (i.e., we did not want to indirectly employ the labels used for training the pretrained encoder).
>
> Notice our method uses single linear layer projections, which strongly restricts the expressiveness of our network if the image encoder is “frozen”. We thus experimented with increasing the depth of these projection networks.
>
> |                   | End-to-end (our setting) |      1 layer  |    3 layers  |    5 layers |
> |---|---|---|---|---|
> | **SIFID** |                     0.05                |            0.26      |      0.14       |   0.17 |
> | **mSIFID**  |                  0.07          |                  0.56      |      0.23    |      0.27 |
> | **Diversity** |                 0.50            |                0.79      |      0.63    |     0.63   |
>
> We accept that if the problem could be learned with a small enough depth, our method would benefit from even faster training, at the cost of enlarging the model (and its consequences on inference time).
> Even though the results are convincing (both visually and quantitatively) in favor of end-to-end training, we prefer not to reject the hypothesis that proper hyper-parameter tuning and perhaps some adaptations could lead to decent results with a frozen backbone.
>
>    Finally, we agree diversity and SIFID metrics have shown some flaws, thus we provided many samples for each model in the supplementary material. Moreover, following the review, we will conduct a user study and share our results.

---

> ### Comment · Reviewer_1LvZ · 2021-08-16
> **Thanks for the additional experiments**
>
> Thanks to the authors for the additional experiments which do indeed seem to indicate that images containing more complex structures also benefit from this approach. I would suggest adding some of these results to the main part of the paper (if possible) as well as your experiments regaring the projections network's depth.
> I have raised my score to 6, assuming that these results and the feedback from the other reviews will be incorporated in some form.

---

> > ### Author Response · Authors · 2021-08-16
> > **Thank you for helping us improve our paper**
> >
> > Thank you for suggesting the new experiments and for approving the results. We will incorporate the results for the new datasets into the main paper and modify the paper as suggested by the four reviewers.

---

### Official Review · Reviewer_mf7L · 2021-07-16

**Rating:** 7
**Confidence:** 4

**Summary:**

The paper aims at improving single-image GAN models by training them on larger datasets. The authors use a meta-learning approach based on the hypernetworks scheme. The model consists of a hyper-generator and a hyper-discriminator. The Hyper-generator network is decomposed into an embedding network that is shared among scales and a linear projection per scale. The hyper-discriminator network is also decomposed into an embedding network and a set of projections. The model is trained with adversarial and reconstruction loss functions. Authors show that the proposed model reduce the training time per image without loss in performance. They also introduce novel
capabilities, such as interpolation and feedforward modeling of novel images.

**Limitations And Societal Impact:**

Yes

**Main Review:**

**Strengths**:


- The paper is well-written, and the proposed approach is simple and intuitive. The authors motivate the problem well.


- Experiments are thorough and demonstrate efficacy of the proposed approach qualitatively and quantitatively. The authors also define new tasks such as interpolation and feedforward modeling which enables new capacities in single-image GAN models.


**Weaknesses**:


- The proposed approach employs hyper-networks in SinGAN models, which is an instance of apply X to Y. Considering that hyper-networks are applied to other generative models and various discriminative models, this makes novelty of the proposed approach moderate. There are existing works such as [A] that use hyper-networks in generative models. There are also recent works such as [B, C, D] which also use hyper-networks for generation (but are not published by the submission deadline for this conference).
However, the good results and improvements of the proposed approach can make up for this.


- The authors note the limitation of their work in terms of model size, convergence, runtime and GPU memory usage in section 6, but do not provide further details. It would be good to have comparisons with SinGAN on this.


[A] HCNAF: Hyper-Conditioned Neural Autoregressive Flow and its Application for Probabilistic Occupancy Map Forecasting; Oh et al.; CVPR 2020


[B] Adversarial Generation of Continuous Images; Skorokhodov et al.; CVPR 2021


[C] pi-GAN: Periodic Implicit Generative Adversarial Networks for 3D-Aware Image Synthesis; Chan et al.; CVPR 2021


[D] HyperVAE: A minimum description length variational hyper-encoding network; Nguyen et al.; 2020

------------------------------------------------------------------------------------------------------------------------------------------------

Update after the rebuttal: I read the rebuttal and other reviews. Overall, I think the paper meets the acceptance criteria for the conference. I keep my score, and I hope the authors clarify concerns of the reviewers in the final version of the paper.

------------------------------------------------------------------------------------------------------------------------------------------------

**Time Spent Reviewing:**

9

---

> ### Author Response · Authors · 2021-08-10
> **Author response**
>
> Thank you for your supportive review.
>
> Regarding the comment on novelty -- while there are multiple ways to apply hypernetworks to the given problem, as we discuss in Sec. 4.2, there are multiple issues that may arise when incorrectly applying hypernetworks in this setting. We theoretically analyze these issues and present a novel solution that is also theoretically grounded (see Prop. 1). Regarding [A-D], while involving hypernetworks, these do not apply internal-learning, thus, do not suffer from generation leakage between samples and the necessity of using a per-instance discriminator, which, as we prove, is a necessity for meta-internal learning.
>
> We attach the requested comparisons with SinGAN -
>
> |              | Time/image | Max # of imgs | Real-time inference |
> |---|---|---|---|
> | **SinGAN** | 50 | 1 | X |
> | **Ours** | <5 | Unlimited | V |
>
> |                 | Main network # params  | Projections # params |  Backbone # params |  GPU |
> |---|---|---|---|---|
> | **SinGAN** | 450K | 0 | 0 | 9GB |
> | **Ours** |  0 | 58M  | 22M | 22GB |

---

> > ### Comment · Reviewer_mf7L · 2021-08-20
> > **Thank you for addressing the comments**
> >
> > Thank you for addressing the comments. The last table basically shows that your model has more capacity than SinGAN. Doesn't this make the comparisons unfair?

---

> > > ### Author Response · Authors · 2021-08-21
> > > **Thank you for the clarification question.**
> > >
> > > Thank you for the clarification question.
> > >
> > > It is important to note that both SinGAN and our primary network, which performs the generation, have exactly the same architecture. Therefore, the capacity of each produced generator (and discriminator) is the same as in SinGAN. We, therefore, compare apples to apples.
> > >
> > > We also note that adding capacity to SinGAN does not improve its performance.
> > >
> > > Unlike SinGAN, our work presents the option to perform meta-learning. The meta-learning network (f) is a ResNet and has a ResNet capacity. Since we trained on unlabeled images (avoiding a pretrained network to ensure fairness) we can benefit from increasing the capacity without paying any price. We note that the same network is used regardless of the size of the training dataset, which is an advantage of ResNets (controlling the effective capacity through the usage of skip connections). We also note that Prop. 1 supports large f networks since with larger f we are able to better approximate the task (lines 69-70 and 179-182) because epsilon becomes smaller in Eq. 18.
> > >
> > > The table also shows that most of the weights are in the projection layer, which is a simple linear transformation that maps the embedding obtained by the ResNet to the weights in the SinGAN architecture.
> > >
> > > Both the ResNet and the linear projection are straightforward choices. Our technical novelty does not focus on the architectural details of the underlying networks but on the proper theoretical way to perform meta-internal learning, showing that hypernetworks are a suitable approach and that it is necessary to employ a hypernetwork for both the generator and discriminator.
> > >
> > > Finally, even disregarding the discussion above, in order to train SinGAN models for 5000 images, for example, one would need 5000 independent models. This would result in more than 2 billion parameters, which is significantly larger than our model that has a fixed capacity that is independent of the number of modeled images.

---

### Official Review · Reviewer_e9qp · 2021-07-17

**Rating:** 7
**Confidence:** 3

**Summary:**

This paper presents a novel single image generation framework that is able to generalize to multi-sample case, which is different from previous work where each model only works on a single sample, leveraging hyper network. The author has provided solid theoretical analysis to support their design. And the effectiveness is also validated by their produced results.

**Limitations And Societal Impact:**

This technique might have a negative societal impact. Some people might be able to leverage this method to produce some fake images based on a given photo. The produced fake images might be harmful.

**Main Review:**

Pros:
1. The paper is well written and easy to follow.
2. The author has provided solid theoretical analysis of the proposed method.
3. The idea of leverage hyper-net to generate the model weights for generator and discriminator is quite novel and intuitive.
4. The paper has made a meaningful progress to generalize the single-sample GAN models to multi-sample case.
5. A collection of variants (shared gen/dis hyper) are also studied to validate the effectiveness of the proposed method.
6. The author has also provided code for reference.

Cons:
1. The author has pointed out the model size might be one of the limitations. I would like to know if there is any failure case existing w.r.t the the generation quality under some circumstances? For example, does the proposed method also works for human face images, etc?

**Time Spent Reviewing:**

2.5

---

> ### Author Response · Authors · 2021-08-10
> **Author response**
>
> Thank you for your supportive review.
>
> Following the review, we have tested the method on 50 randomly sampled face images from the CelebA dataset.
> Since our work is an extension of the single-image framework, our generation is still based on patches within each image. In the case of facial datasets, as far as we are aware of, the best case one can hope for is some slight facial characteristics changes, for example varying the width of the nose, the jawline, and the distance between the eyes. The latter is achievable by our method in a coherent manner by training with a smaller (spatially) initial noise.
>
> We attach side-by-side results here  [Link](https://imgur.com/a/fhSCQXE) along with the baselines, where we used the same initial noise size for all of the methods to allow for a fair comparison.
> As can be seen, our method outperforms the baselines for generating faces and takes approximately x10 and x5 less time to train (by image) than SinGAN and ConSinGAN respectively.
> Even though the results obtained are encouraging in the context of the single-image generation framework, we are aware that these results are non-comparable to classic face generation (by standard GAN and Flow-based models), and thus, can be considered as a limitation. This will be addressed in the next version of the paper.
>
> Regarding other types of visual data, we refer the reviewer to [Link 1](https://imgur.com/a/0xpMcVu) and [Link 2](https://imgur.com/a/TYNQfSc) that show our results on images from the LSUN dataset (performed at the request of Reviewer 1LvZ)

---

### Decision · Program_Chairs · 2021-09-27

**Decision:**

Accept (Poster)

**Comment:**

Meta-review of "Meta Internal Learning"

This paper proposes a framework for single image generation from the perspective of meta-learning on larger datasets. The method uses hypernetworks as the generator and discriminator so that the weights of these networks can be conditionally adapted from the single image from a set of projections, and can be trained with adversarial and reconstruction losses. They discuss capabilities of the method such as interpolation and modeling of novel images.

Most reviewers agree that this paper is well written, easy to follow, includes solid analysis of the proposed methods, and also generally agree with the novelty. The fast inference can find important uses in real-world applications. The weaknesses are highlighted by the authors and discussed in detail with reviewers, and the authors have included several comparisons to previous work. As suggested by reviewer FnNF (who had increased their score), the authors are strongly encouraged to better reorganize the manuscript in the camera ready version, in which the authors agreed to do so.

For all these reasons stated, I recommend acceptance of this work.